# SAQ training on sprint, change-of-direction speed, and agility in U-20 female football players

**Young-Soo Lee[1], Dayoung Lee[2]\*, Na Young Ahn[3]**

**1** Department of Physical Education, Sejong University, Seoul, South Korea, **2** Department of Sport Science, Kangwon National University, Chuncheon, South Korea, **3** Department of Sport and Event Management, Bournemouth University, Poole, United Kingdom

\* ldy227@kangwon.ac.kr

## Abstract

The purpose of the study was to investigate the effects of an 8-week speed, agility, and quickness (SAQ) training on performance changes in linear sprint speed, change-of-direction (COD) speed, and reactive agility of U-20 female football players. Nineteen female football players randomly served as either experimental ($n = 9$) or control groups ($n = 10$). The players were tested for physical fitness tests: linear sprint speed including both short and long distances (5- and 10-m sprints without a ball and 20- and 30-m sprints with and without dribbling), COD speed (arrowhead agility test with and without dribbling a ball, Southeast Missouri [SEMO] agility test, and 22-m repeated slalom dribbling test), and reactive agility. Significant group × time interactions were observed for sprint over long distances and COD speed but not for short sprint and reactive agility performances. Paired t-tests revealed considerable improvements in all performances from the pre-test to post-test for the SAQ group, except for the arrowhead agility (left; $p = .07$). The control group only exhibited significant improvements in 10-m sprint performance after general football training. Eight weeks of SAQ training were effective at enhancing acceleration, maximum sprint speed, and agility performances amongst highly trained U-20 female football players.

## Introduction

Football is a physically intense sport with intermittent and rapid actions. For example, professional players engage in 5–6 training sessions and play 1–2 games a week during a season [1]. During a game, players run approximately 10 km involving varying levels of intensity and sprint every 70 seconds at a high-intensity [2]. They also constantly carry out dynamic and brief actions (e.g., passing, kicking, trapping, dribbling, tackling) and without a ball (e.g., high-speed running, accelerations, decelerations, and change-of-direction [COD]) [3–5]. Given the demands, players are required to possess multifaceted physical features. These include muscular strength and endurance, repeated sprint ability, sprinting speed, agility, to name a few, to perform high intensity actions with and without possession of a ball throughout a game [6–8].

**Data Availability Statement:** All relevant data are within the paper and its Supporting information files.

**Funding:** The authors received no specific funding for this work.

**Competing interests:** The authors have declared that no competing interests exist.

Researchers have emphasised the significance of players' high-speed movements, characterised by constant changes in velocity and direction, which account for less than 12% of the overall football performance; these movements in turn greatly influence the outcome of a game [2,9]. Acceleration and deceleration speed, maximum sprint speed, COD speed, and reactive agility are the contributing factors to high-intensity performances during a game. Acceleration and deceleration speed involves sprinting momentum over linear course (10- to 30-m), where the maximum acceleration speed can occur during the early stages of sprint start (0- to 5-m). COD speed is another aspect of agility that requires players to coordinate pre-planned and sports-specific movements maintaining body control [10]. While COD speed relies on players' physical capacity to change directions, agility requires players to rapidly process and react to diverse stimuli, performing perceptual and cognitive processing [11]. Hence, these high-speed actions are independent qualities and unrelated to each other [10,12–13]. For instance, some players excel in sprinting may have weaknesses in decision-making or precise reaction to stimuli, whereas others may maintain maximum sprint speed and control during sudden changes in direction, accelerations, and decelerations [11]. As a result, coaches and researchers strive to develop appropriate training regimes that strengthen the incorporated effects of acceleration and deceleration speed, COD speed, and agility. This approach aims to achieve high levels of physical performance in complex and dynamic movements [14].

SAQ training was specifically designed to enhance players' COD speed and agility performances and improve high-speed running and sprint capabilities during a game [15]. Combining a range of movement tasks, including lateral shuffles, forward and backward running, and ladder drills, SAQ training enables players to enhance their rapid transitions, motor coordination, and reaction time. In other words, SAQ training focuses on performing movement tasks at a high rate in short-time (quickness) incorporated both straight-line speed and multidirectional sprints (COD speed) across various distances to enhance maneuverability [16]. Researchers have thus integrated SAQ training into football-specific exercises and conditioning programmes and examined its effects on physical performance of football players [7,8,12,17–20]. In their study of young football players, Jovanovic and associates [12] found that an 8-week SAQ training programmes implemented during the season resulted in improved acceleration and short sprint performance compared to the group without the intervention. Milanović and associates [7] also found that a 12-week combined SAQ and flexibility training programms led to significant enhancements in speed, especially at short distances of 5 and 10 m, compared to a group that underwent regular training only. Furthermore, Formenti and colleagues stressed that SAQ training with a combination of closed (sprinting and COD) and open (balance and reactive movements) drills is effective when both football-specific and non-sport specific stimuli are incorporated, considering youth players' perceptual response [21]. Finally, Azmi and Kusnani [22] trained football players for 8 weeks and concluded that the SAQ training method effectively improved sprint, agility, reaction time, and power in football players.

Although Yap and colleagues [23] have highlighted the importance of developing physical fitness, such as sprint speed and agility, for both female and male players to excel in high-intensity actions during a game, much of the research in this area has focused on male football players. To our knowledge, only a few researchers have empirically tested the propositions of Yap and Colleagues [23] showing positive outcomes of the SAQ exercises among female football players, both in youth [24] and adult [25] categories. Polman and associates [25] highlighted as much in their study of female football players, arguing that SAQ training using speed-ladder, hurdles, and reaction balls improved the physical and biomechanical characteristics of female football players. Considering physical qualities are cultivated from a young age,

such information holds particular significance for young female players aiming to improve sustainable high-intensity running over long distances and COD speed. Further, despite the importance of and benefits from SAQ training, little is known about its effects in highly trained female youth football players, with an 8-week intervention [see 26 for classification of athletes]. Previous studies on male football players have reported notable improvements in short sprint performance and COD ability throughout the 8 weeks of SAQ training [12,22]. Similarly, Mathisen and Svein [24] found that SAQ training over the course of 8 weeks improved linear sprint speed and agility performance in female youth football players. When implementing a training method, 5–6 weeks are enough to induce significant physiological and neuromuscular adaptations in highly trained football players during pre-season and in-season [19,27]. Given the limited empirical evidence, the aim of the study was to investigate the effects of an 8-week SAQ training in the selected physical capabilities of female football players. We hypothesised that implementing the SAQ training over 8 weeks will improve linear sprint speed, COD speed, and reactive agility.

## Materials and methods

### Participants

Nineteen female football players (age = 18.89 ± 0.80 yr; height = 164.54 ± 4.11 cm; body mass = 58.28 ± 6.57 kg; and body mass index or BMI = 21.52 ± 3.46 kg/m2) competing at the national level volunteered to participate in the study. Prior to conducting the study, a power analysis was performed using G*Power software [28]. The analysis indicated that for a two-tailed t-test with moderate effect size of .3, power of 80%., and alpha level at .05, a minimum sample size of 176 per group was recommended. However, due to the scarcity of highly trained players, our participant pool for the study was smaller. Further, while an experimental design with equivalent group is ideal to compare pre-test and post-test effects, practical difficulties led us utilise a nonequivalent control group, with the exception of one player in the study. All players were members of a single regional intercollegiate football team registered at a local football association and had at least 5 years of football experience. None of the players had had any serious injuries 6 months prior to the initial testing, and none had ever participated in SAQ training. Based on the uneven and even numbers of player jerseys, each player was randomly assigned to either an SAQ training group (SAQ; n = 9; age = 19 ± 0.86 yr; height = 165.62 ± 4.10 cm; body mass = 59.91 ± 7.78kg; and BMI = 21.84 ± 4.62 kg/m2) or general training group (GTG; n = 10; age = 18.80 ± 0.78 yr; height = 163.58 ± 4.09 cm; body mass = 56.83 ± 5.26kg; and BMI = 21.23 ± 2.19 kg/m2). No large changes in body mass, height, or injuries were observed over the duration of the study. Table 2 presents the descriptive data of the SAQ and GTG groups.

The study was approved by the ethics committee of Sejong University to ensure the safety and privacy of human subjects. Before providing written consent, all players were informed of the potential risks and benefits of participation and that they could withdraw from the study at any time.

### Design and procedures

In a two-group, randomised controlled design, the effects of an 8-week SAQ training on acceleration, linear sprint speed, COD speed, and reactive agility of female football players were assessed by comparing two experimental conditions: an intervention and control group. Performance was evaluated using the following tests: 5- and 10-m sprints without a ball and 20- and 30-m sprints with and without dribbling a ball for speed parameters; arrowhead agility tests with and without dribbling a ball, the Southeast Missouri (SEMO) agility test, and 22-m

repeated slalom dribble test for COD speed parameters; and reaction agility test for agility assessment.

All experimental procedures took place for 8 consecutive weeks, from February (the pre-season) to April (the beginning of the in-season) in 2018. All physical conditioning and tests took place on a full-size (90 x 120 m) outdoor artificial turf pitch between 3:30 PM to 5:30 PM and were guided by experienced football coaches certified by a continental football association. All players were familiarised with the different training protocols and testing procedures prior to the initiation of the study and wore football boots throughout the training and testing sessions. To ensure consistency, all players were instructed to maintain normal dietary intakes consisting of a standardised breakfast, lunch, and dinner, as well as their regular lifestyles during the study. This approach aimed to minimise any uncontrolled impacts on the results.

## Training

Over the 8 weeks, all players performed either the SAQ training protocol or substitute drills involving ball skills, such as dribbling, passing, and shooting. These trainings occurred 3 times a week for 40 minutes per session, with at least 24 hours of recovery between each conditioning session, and the sessions were scheduled 2–3 days apart. Both groups of players underwent the same volume of a specific physical conditioning (16 hours) but at a different level of intensity. The experimental group performed the SAQ training with a progressive intensity from 80% to 100%, whereas the control group exercised normal football drills at various intensities within the range of 80% to 100%. The researchers ensured that all players reached at least 80% of intensity during conditioning sessions. Following this, both groups maintained the same volume and intensity for the remaining practice. Each training session lasted up to 120 minutes and consisted of the programmed conditioning training, technical and tactical drills, and small-sided games or match-play. Regarding the intensity of the training sessions, we consistently monitored the participants' heart rate (HR) and rating of perceived exertion (RPE). However, we did not document these records as they were not within the scope of the study. The overview of the 8-week SAQ training, along with the programmes for the control group, is presented in Table 1.

## Testing protocol

Performance testing was conducted twice, before and after the 8-week SAQ training during regular training hours, with a 24-hour recovery between the training and testing. Prior to testing, all players performed a standard warm-up, consisting of 10–15 minutes of jogging, dynamic stretching with progressive speed runs, and ball contact exercises. The measurements included acceleration and maximum sprint speed, COD speed, and reactive agility. Each player was instructed to give their maximal effort during performance testing. All players completed a maximum of 2 trials for every measure to become familiar themselves with a battery of physical fitness tests, with a minimum of 5 minutes of recovery time between each test. At the end of testing, all players performed the same cool-down for 10–15 minutes involving low-intensity running and static stretching. On the first day of testing, the body mass and height of each player was measured using a stadiometer and electronic scale (FA-94H, Fanics and DB-150A, CAS, South Korea, respectively). The BMI was also calculated as weight/height squared (kg·m–2), and there were no significant differences between the groups on anthropometric data.

**Speed testing.** Speed refers to one's ability to move a body as fast as possible, which is often measured by linear sprinting speed [29]. Whilst 5- to 10-m sprints are considered as

**Table 1. An overview of the 8-week SAQ and general training programmes.**

| Weeks | SAQ Training | General Training |
|---|---|---|
| 1–2 Intensity 80% | Speed<br> Wall Drill 3 Count–A Skip, A Run, and B Run<br>COD Speed<br> Hop Scotch<br> Agility Ladder Drill<br>Reactive Agility<br> Reaction Ball Drop | Passing<br> With inside and laces of a foot or head<br> On and partly above the ground<br> Driven and lofted passes (with a spin)<br> In pairs or groups of 3–6<br> 1–3 touches<br> Moving to receive the second ball<br>Ball Controlling |
| 3–4 Intensity 90% | Speed<br> Speed Ladder Drill<br> Speed Training Band<br>COD speed<br> Agility Ladder Drill<br>Reactive Agility<br> Mirror Game | With inside, outside, and sole of a foot for receiving ground balls<br> With feet, thigh, chest, and head<br> With a 90˚ or 180˚ turn<br> In pairs or groups of 3–4<br> In a fluid motion<br>Dribbling<br> With inside and outside of a foot<br> Fake moves–Step-Over, Scissors, etc. |
| 5–6 Intensity 90% | Speed<br> Speed Hurdle Drill<br>COD speed<br> Slalom Sprint<br> Agility Hurdle Drill<br>Reactive Agility<br> Reaction Training | In a different pace<br> In a different direction–forward, etc.<br> Turning and cutting<br> At speed<br>Kicking<br> With inside, outside, laces of a foot<br> Chipping and crossing<br> Short and long distances<br> In pairs or groups of 3–4 |
| 7–8 Intensity 100% | Speed<br> Sprint<br> Shuttle Run<br>COD Speed<br> Illinois Agility Run<br> T-Drill<br>Reactive Agility<br> Reaction Sprint | Shooting<br> With inside and outside of a foot or head<br> With instep drive on a half or side volley<br> In the near, middle, and far post<br> From distance and difficult positions<br> Quick turn towards a goal<br> Solo and combination play to finish<br> 1–2 touches |

**Note**: SAQ = speed, agility, and quickness. COD = change-of-direction.

indicators of acceleration speed [8,19], sprints over 10 m are used to assess high-running velocity. In football, where the average sprint distance is around 20 m and rarely exceeds 30 m [30], maximum sprint speed was measured by 20- and 30-m sprints [31]. To evaluate players' sprinting capabilities, all players performed 12 straight-line sprints: two trials over 5-, 10-, 20-, and 30-m without dribbling a ball, and another 2 trials while dribbling a ball over each distance of 20 and 30 m.

*5-m and 10-m Sprint Tests.* All players began by standing upright, with the toe of the front foot positioned right behind the start line. The players sprinted as quickly as possible through the end line and rested 2 minutes in between. The same starting and recovery strategies were used for subsequent analysis. Sprint times were recorded by the first author using timing gates placed on the start and finish lines in units of 0.01 seconds (Weltek, TK-9920, Star Sports, South Korea), and the best record for each test was attained for data analysis.

*20-m and 30-m Sprint Tests.* All players completed two sets of maximal straight sprints without a ball over 20- and 30-m, with the same starting and recovery strategies as employed in the short sprint tests. The same photoelectric cells were used to measure maximum sprint speed, and the best result for each test was retained for further analysis.

*20-m and 30-m Sprint Tests with a Ball*. The dribbling speed test reflects the entities of maximum sprint speed and dribbling skill under time pressure. All players were instructed to dribble a standard ball to the finish point, while maintaining maximum velocity throughout. The same starting, recovery, and recording strategies were adopted from the previous sprint tests. To ensure validity, a trial was considered valid when the players completed at least 4 touches for 20-m and 6 touches for 30-m sprint [31]. Otherwise, the trial was discarded and reattempted, and the fastest time was taken for further analysis.

**COD speed and agility testing.** Agility is defined as one's ability to quickly change direction in response to a stimulus (reactive agility), while COD speed coordinates pre-planned, sports-specific movements, maintaining body control [10]. Following the speed testing, all players underwent the arrowhead agility test, both with and without dribbling, as well as the SEMO agility test and the 22-m repeated slalom dribbling test, which are highly reliable and widely used to measure COD speed [32,33]. When testing, the same procedure was used for the starting position and recovery phase as in the speed testing (i.e., a 2-min rest interval between 2 trials and a 5-min between each test), with the exception of the reactive agility test.

*COD Speed Tests*. To measure COD speed without a ball, all players were instructed to sprint and perform COD movements maximally through a certain sequence of cones as depicted in Figs 1 and 2.

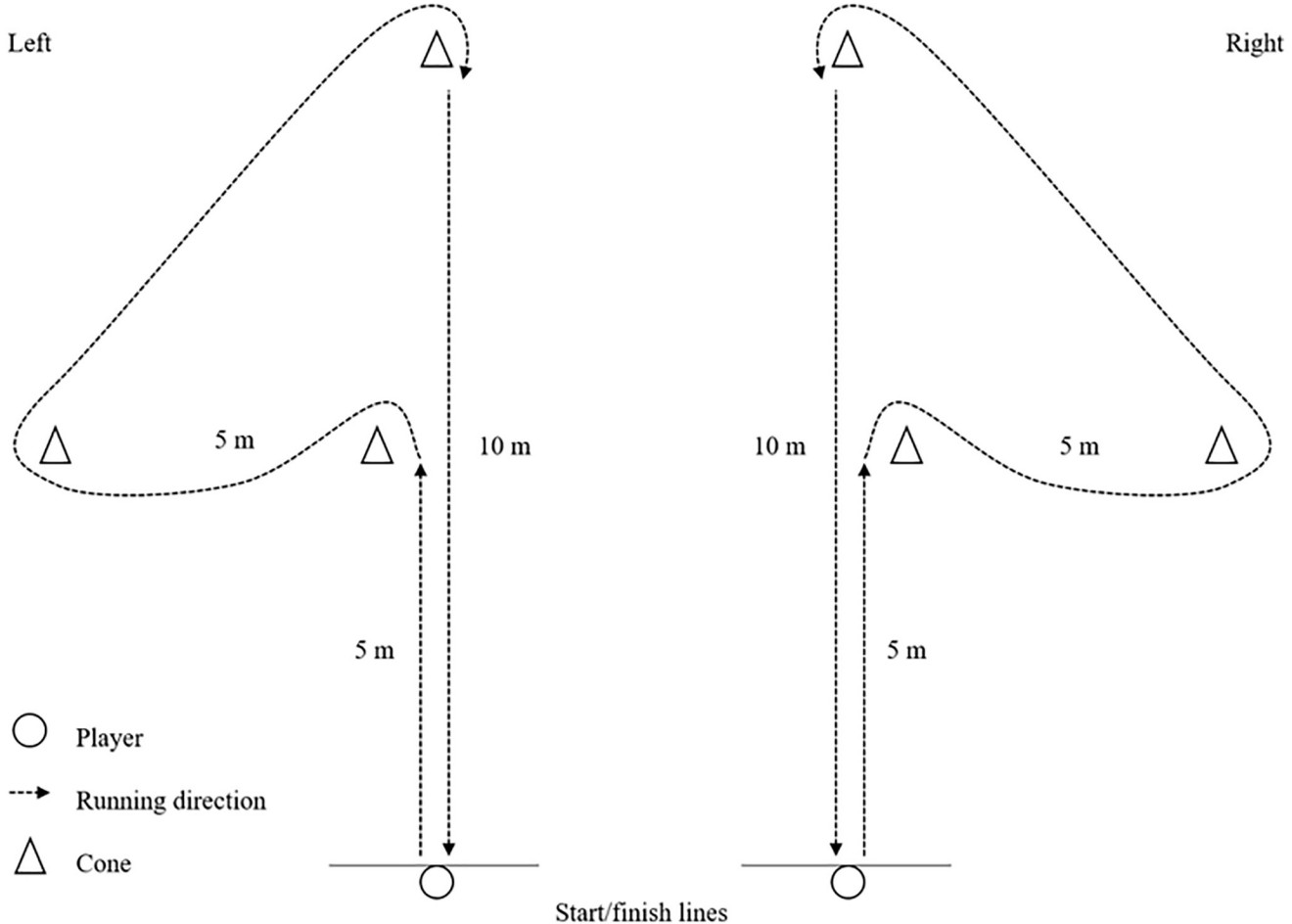

**Fig 1. Arrowhead agility test.**

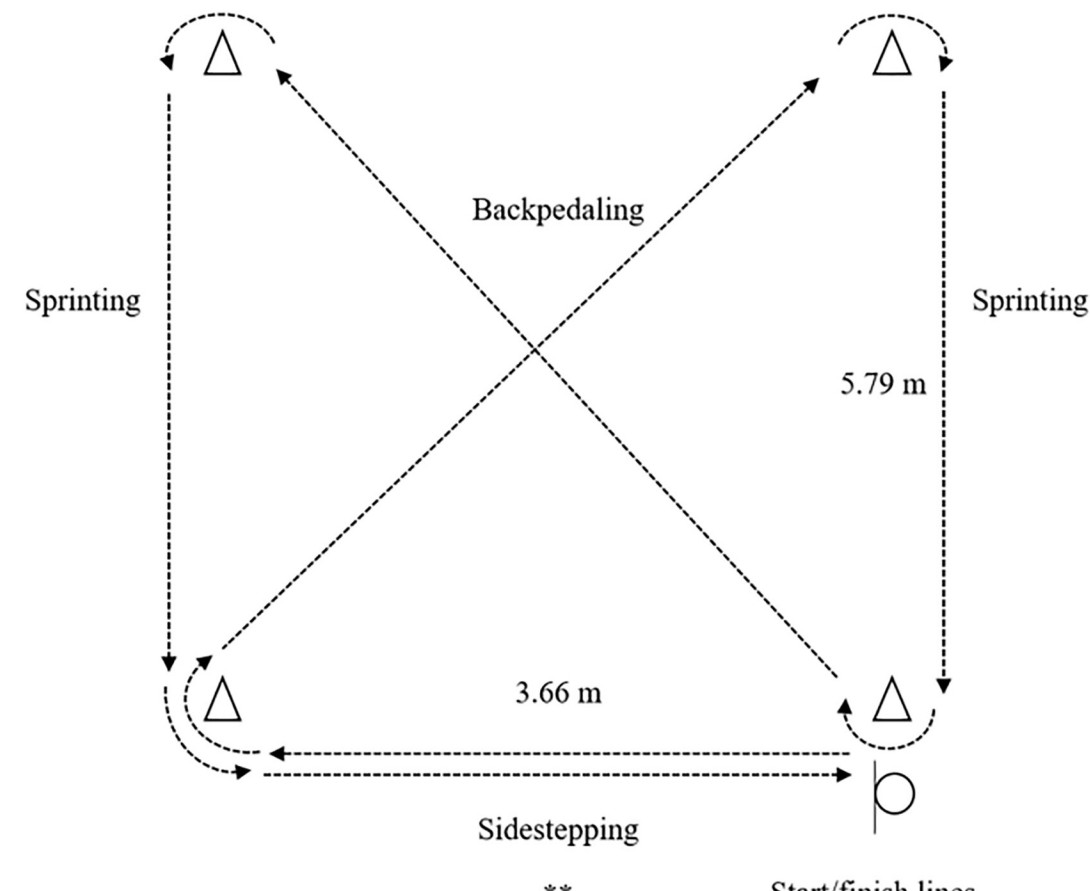

**Fig 2. SEMO agility test.**

The players completed the arrowhead agility test twice, both in the left and right directions [see 33]. For the SEMO agility test, the players were required to perform forward sprinting, lateral shuffling, and diagonal backpedaling while turning around the cones [see 34]. Six trials without dribble were completed, and the attempt was considered invalid if the players stepped on, moved over, or knocked down a cone. Each trial was timed by the first author using a handheld stopwatch with a precision of 0.01 seconds [35], and the fastest times were recorded for data analysis.

*COD Speed Tests with a Ball.* The dribbling agility test evaluates both COD speed and dribbling performance under control. Each player performed 4 trials of the arrowhead agility test, dribbling a ball through each direction (Fig 1). For the 22-m repeated slalom dribble test [see 32 for details], the players were instructed to dribble slalom as quickly as possible through a series of 14 cones placed 2 m apart, as shown in Fig 3.

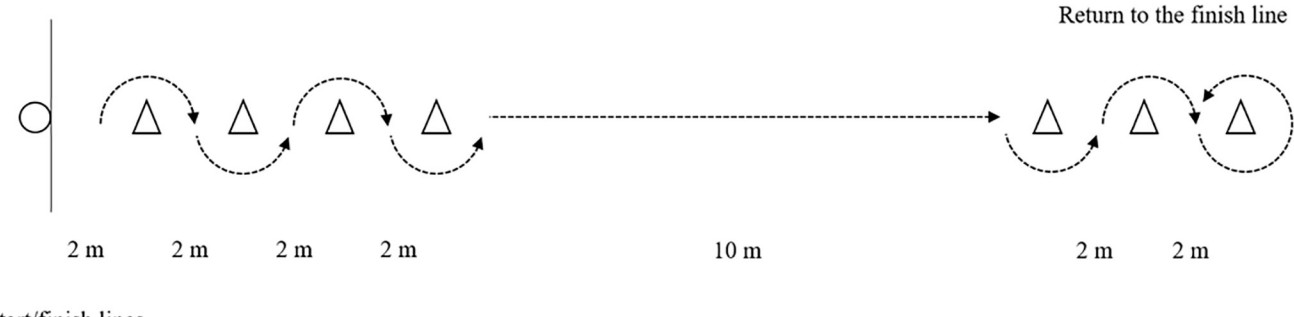

2 m   2 m   2 m   2 m   10 m   2 m   2 m

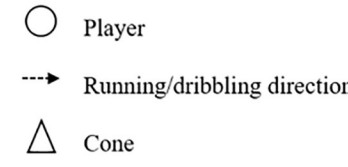

Start/finish lines

○ Player

---▶ Running/dribbling direction

△ Cone

**Fig 3. 22-m slalom repeated dribble test.**

Upon reaching the last cone, the players made a 180º turn and dribbled back through the cones to the starting line. The trial was disregarded and reattempted if the players disrupted a cone. Each player completed 6 trials, and all tests were conducted by the same rater using a handheld stopwatch, and the best time was attained for further analysis.

*Reactive Agility Testing.* Reactive agility refers to unplanned, dynamic movement patterns in response to sport-specific cues [20]. These tasks involve open skill agility, requiring players to quickly and efficiently read and react in a game situation, to take an appropriate action based on stimuli [11]. Following the completion of COD speed testing, all players were asked to sprint as fast as possible to assess how promptly they produce unplanned directional changes while processing sport-specific cues.

The testing protocol was adopted from previous research, as depicted in Fig 4 [see 20].

On command, the players moved 5 m forward, passing through a trigger gate at a maximal speed. Upon passing through the trigger gate, they changed direction either to the left or right side based on cues provided by the researcher. The players then performed a 45º cut and sprinted an additional 5 m through a target gate while still maintaining their full speed. Each player completed 2 trials, with a 3-minute rest between each attempt. The same method was used for the starting position as used in the speed testing. Times from the start to finish lines were measured by the same researcher using a handheld stopwatch, and the best result was used for data analysis.

## Statistical analyses

Data are presented as mean ± standard deviation. The data were stored in Microsoft Excel and analysed using the R language (version 4.2.3 by R Core Team, Vienna, Austria) with R Studio (version 2023.09.0+463; R Studio PBC, Boston, USA). Our final dataset is presented in S1 File. The assumptions of normality and homogeneity of variances were checked by Shapiro-Wilk and Levene's test on the residual terms of the change scores, while the sphericity assumption was not applicable because there were only two levels of the within-subjects factor (i.e., time). For the inferential analysis, two-way (group × time) mixed factorial analyses of variance

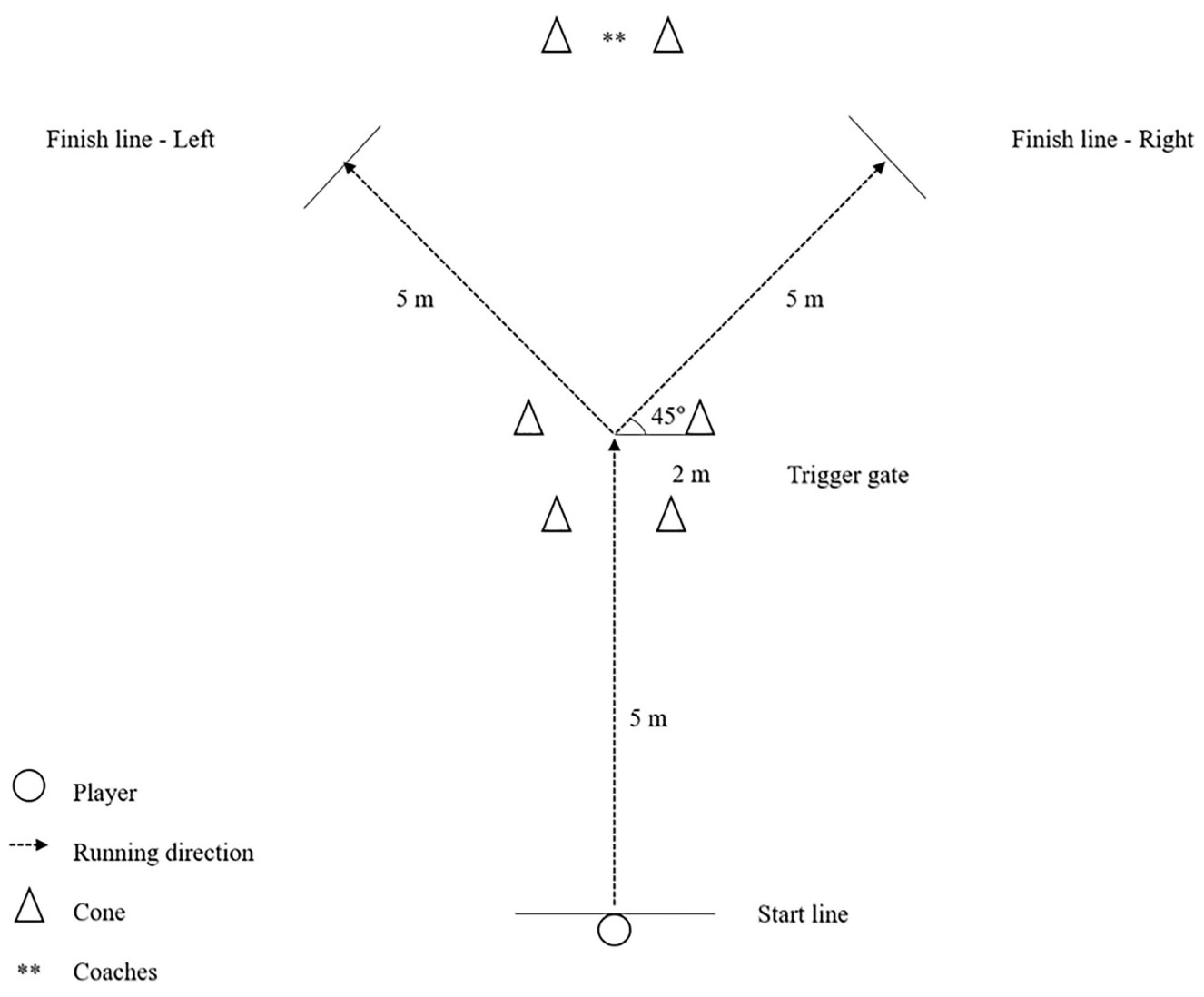

**Fig 4. Reaction agility test.**

(ANOVA) were employed to investigate both the interactions and main effects of training and time. In case of a significant interaction, pairwise comparisons were not applied because the factors entailed only two levels. Instead, independent or paired t-tests were conducted to determine any significant differences between the two groups or test occasions within each group. For specific conditions that did not satisfy the homogeneity of variance, the Mann-Whitney test was used to assess significant differences between the groups during the post-training assessment. Cohen's d and partial eta squared ($\eta$2) were used to determine the magnitude of mean differences in t-tests and ANOVAs, respectively. Effect sizes were evaluated based on the following criteria [36]: values of .2, .5, and .8 for small, medium, and large effects of $d$, and .01, .06, and .14 considered as the cut-off for small, medium, and large values of $\eta$2, respectively. Statistical significance was set at $p < .05$. Table 2 presents the summary statistics for all variables used in the study.

**Table 2. Changes in measure with confidence intervals and correlation matrix.**

| variables | Change in measure | | | 1 | 2 | 3 | 4 | 5 | 6 | 7 | 8 | 9 | 10 | 11 | 12 | 13 |
|---|---|---|---|---|---|---|---|---|---|---|---|---|---|---|---|---|
| | SAQ (n = 9) [LL, UL] | GTG (n = 10) [LL, UL] | Mean Difference [LL, UL] | | | | | | | | | | | | | |
| Age (y) | 19.00 ± .86 | 18.80 ± .78 | | | | | | | | | | | | | | |
| Height (cm) | 165.62 ± 4.10 | 163.58 ± 4.09 | | | | | | | | | | | | | | |
| Body mass (kg) | 59.91 ± 7.78 | 56.83 ± 5.26 | | | | | | | | | | | | | | |
| BMI (kg/m$^{-2}$) | 21.8 ± 4.62 | 21.2 ± 2.19 | | | | | | | | | | | | | | |
| 1 5-m sprint | -.10 ± .07 [-.16, -.05] | -.03 ± .10 [-.11, .04] | -.06 ± .09 [-11, -.02] | — | | | | | | | | | | | | |
| 2 10-m sprint | -.16 ± .09 [-.23, -.08] | -.09 ± .10 [-.16, -.01] | -.12 ± .10 [-.17, -.07] | -.09 | — | | | | | | | | | | | |
| 3 20 m sprint | -.16 ± .15 [-.28, -.04] | .06 ± .12 [-.02, .16] | -.04 ± .18 [-.12, .04] | .23 | .15 | — | | | | | | | | | | |
| 4 30-m sprint | -.25 ± .15 [-.37, -.14] | .02 ± .21 [-.12, .17] | -.10 ± .23 [-.21, .01] | .68** | .39 | .41 | — | | | | | | | | | |
| 5 20-m dribble | -.21 ± .14 [-.32, -.10] | .01 ± .21 [-.14, .15] | -.09 ± .20 [-.19, .01] | .37 | -.02 | .43 | .40 | — | | | | | | | | |
| 6 30-m dribble | -.24 ± .24 [-.43, -.05] | .08 ± .24 [-.09, .25] | -.07 ± .29 [-.21, .07] | -.15 | .57* | .24 | .16 | .17 | — | | | | | | | |
| 7 Arrowhead agility (L) | -.11 ± .16 [-.24, .01] | .26 ± .27 [.06, .46] | .08 ± .29 [-.05, .22] | .37 | -.10 | .72*** | .42 | .67** | .02 | — | | | | | | |
| 8 Arrowhead agility (R) | -.21 ± .13 [-.31, -.11] | .15 ± .27 [-.03, .35] | -.02 ± .28 [-.15, .11] | .57* | -.07 | .47* | .45 | .56* | .13 | .80*** | — | | | | | |
| 9 SEMO agility | -.64 ± .37 [-.93, -.35] | .56 ± .47 [.22, .90] | -.01 ± .74 [-.36, .35] | .49* | .02 | .64** | .61** | .48* | .27 | .74*** | .77*** | — | | | | |
| 10 Arrowhead dribble (L) | -.25 ± .24 [-.44, -.06] | .49 ± .46 [.15, .82] | .13 ± .53 [-.11, .39] | .17 | .36 | .42 | .29 | .54* | .55* | .34 | .45* | .54* | — | | | |
| 11 Arrowhead dribble (R) | -.25 ± .17 [-.38, -.11] | .53 ± .46 [.20, .86] | .16 ± .53 [-.09, .42] | .36 | .20 | .43 | .54* | .37 | .22 | .65** | .66** | .55* | .35 | — | | |
| 12 22-m slalom dribble | -1.82 ± .80 [-2.44, -1.20] | -.01 ± .71 [-.51, .50] | -.86 ± 1.19 [-1.44, -.29] | .44 | .20 | .39 | .55* | .48* | .47* | .43 | .57* | .58** | .54* | .68** | — | |
| 13 Reactive agility | -.13 ± .09 [-.20, -.06] | -.05 ± .22 [-.22, .10] | -.09 ± .17 [-.18, -.01] | -.06 | -.04 | .36 | -.12 | .11 | .12 | .34 | .34 | .18 | -.01 | .16 | .22 | — |

Note: SAQ = speed, agility, and quickness. GTG = general training group. *n* = sample size. LL = lower limit. UL = upper limit. BMI = body mass index. COD = change-of-direction. SEMO = Southeast Missouri. L = left. R = right. The values in the brackets represent the 95% confidence interval (CI) for each change in measure for both groups and mean difference between the groups.

*$p < .05$.

**$p < .01$.

***$p < .001$ (two-tailed).

## Results

With the normality assumption confirmed, the independent t-tests were conducted to compare the means of the pre-test between the two groups. The results showed that the groups were equal (all $p > .05$), except for the 22-m repeated dribble test. To assess the training effect of this variable, the Mann-Whitney test was used to evaluate the differences between the groups at the post-test.

**Table 3. Results of two-way mixed analyses of variances.**

| DVs | 2 × 2 mixed ANOVA $(df_1 = 1; df_2 = 17)$ | | | | | |
| --- | --- | --- | --- | --- | --- | --- |
| | Main Effects | | | | Interaction | |
| | Group | | Time | | Group × Time | |
| | $F$ | $p$(partial $\eta^2$) | $F$ | $p$(partial $\eta^2$) | $F$ | $p$(partial$\eta^2$) |
| Speed | | | | | | |
| 5-m sprint | .04 | .83(.01) | 11.06 | .01(.39)** | 2.93 | .10(.14) |
| 10-m sprint | 1.60 | .22(.08) | 30.88 | .01(.64)*** | 2.15 | .16(.11) |
| 20 m sprint | 0.47 | .50(.02) | 2.23 | .15(.11) | 13.45 | .01(.44)** |
| 30-m sprint | 1.82 | .19(.09) | 7.30 | .01(.30)* | 11.25 | .01(.39)** |
| 20-m dribble | 3.06 | .09(.15) | 6.00 | .02(.26)* | 6.82 | .01(.28)* |
| 30-m dribble | .54 | .47(.03) | 1.95 | .18(.10) | 8.33 | .01(.32)* |
| COD Speed | | | | | | |
| Arrowhead agility (L) | 1.00 | .33(.05) | 1.95 | .18(.10) | 12.56 | .01(.42)** |
| Arrowhead agility (R) | .01 | .91(.01) | .37 | .54(.02) | 14.27 | .01(.45)** |
| SEMO agility | .03 | .85(.01) | .16 | .69(.01) | 37.78 | .01(.69)*** |
| Arrowhead dribble (L) | 8.85 | .01(.34)** | 1.84 | .19(.09) | 18.47 | .01(.52)*** |
| Arrowhead dribble (R) | .15 | .69(.01) | 3.00 | .10(.15) | 23.12 | .01(.57)*** |
| 22-m slalom dribble | .06 | .80(.01) | 27.64 | .01(.61)*** | 27.56 | .01(.61)*** |
| Reactive Agility | | | | | | |
| Reactive agility | 1.98 | .17(.10) | 5.69 | .02(.25)* | .89 | .35(.05) |

**Note**: ANOVA = analysis of variance. $df$ = degrees of freedom. DV = dependent variable. COD = change-of-direction. SEMO = Southeast Missouri. L = left. R = right.

*$p < .05$.

**$p < 0.01$.

***$p < .001$.

Times are expresses in second.

## Speed

All group × time interactions were statistically significant for longer linear sprints, with large effects on speed (Table 3 and Figs 5–8).

The main effect of time was also statistically significant for 5-, 10-, and 30-m sprints without a ball, as well as 20-m sprints with dribbling, with large effects. However, there was no significant effect on performance based on the group. Paired t-tests demonstrated significant improvements in all speed measures in the SAQ group after 8 weeks of training. On the other hand, the control group only showed greater reductions in 10-m sprint time after regular football training [see Table 4]. Further analysis showed that the SAQ group had significantly better performance in all speed measures at the post-testing, except for 5- and 10-m sprints, with large effects ranging from 1.05 to 1.42, compared with the control group.

## COD speed

The results of the mixed-model ANOVA revealed that the group × time interactions were significant for all other measures, with large effects (Table 3 and Figs 9–13).

Subsequent analysis using paired t-tests revealed significant improvements in all COD speed measures, except for the arrowhead agility (left) in the SAQ group. In contrast, the control group showed significant increases in testing time in the arrowhead agility (left), the

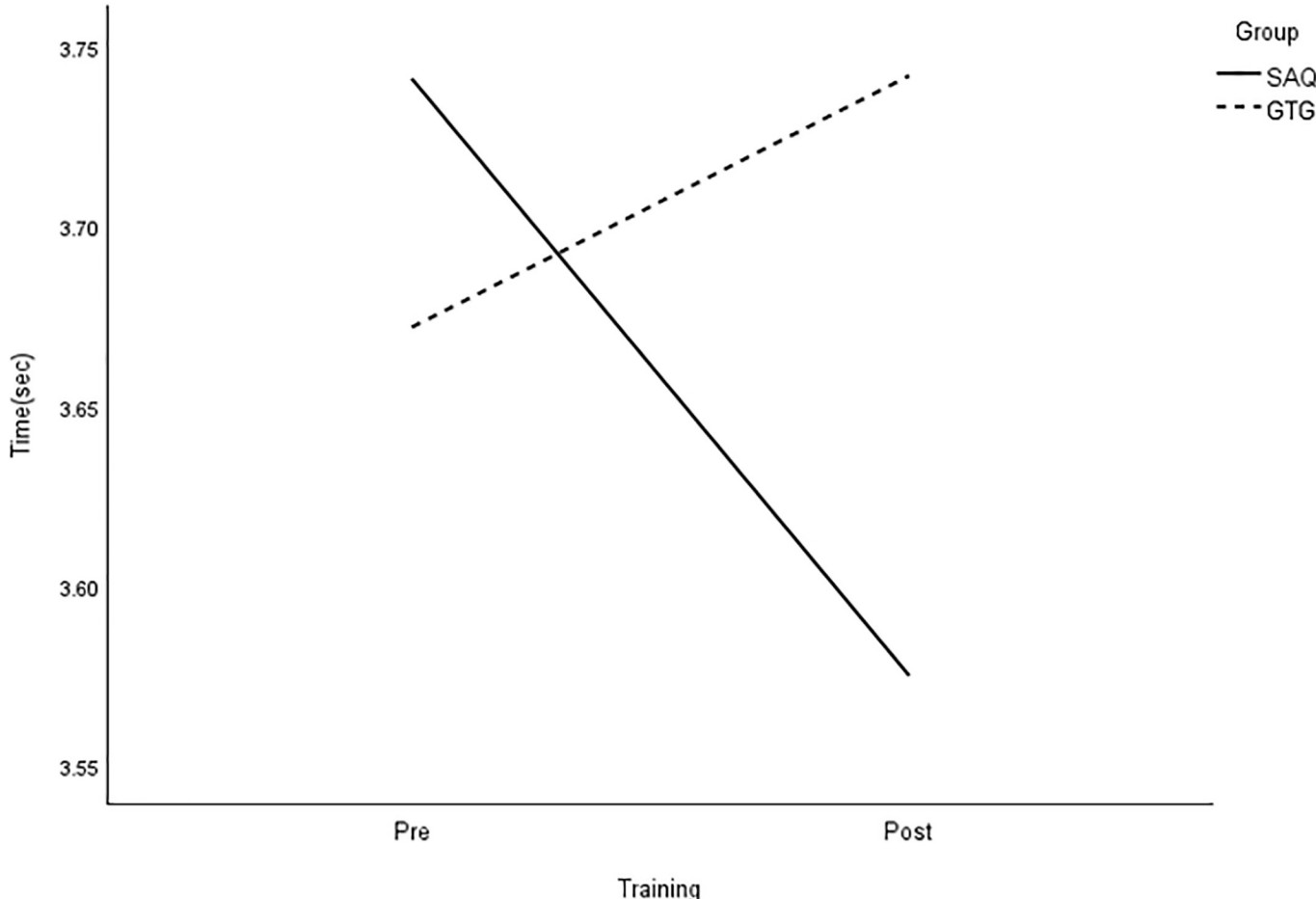

**Fig 5. Interaction plot for 20-m sprint.** *Note*: SAQ = speed, agility, and quickness training group. GTG = general training group. The results of a two-way mixed analysis of variance (ANOVA) indicated a significant group × time interaction for 20-m sprint ($p < .01$) during the 8-week training period, while no main effects were found. Post hot analysis showed a greater improvement in the SAQ group compared to the GTG group ($p < .01$).

SEMO agility, and arrowhead dribbling tests for both sides, with large effects (Table 4). Finally, the independent t-tests indicated that performance improved in the SEMO agility and the arrowhead agility with dribbling (left) in the SAQ group at the post-test. The Mann-Whitney test also revealed that the post-test scores of the 22-m slalom dribbling were significantly lower in the SAQ group compared to the control group ($U = 16$, $z = $ -2.36, $p = 0.01$; Table 4).

### Reactive agility

A 2 × 2 mixed ANOVA was conducted to analyse the reactive agility test results. The analysis revealed that the group × time interaction was not significant (Table 3), while there was a significant main effect of time, with large effects, but not for the effect for group. A paired t-test was run to further explore the differences between the two testing sessions for each group. Significant enhancements were apparent in the reactive agility test for the SAQ group after training, while no significant differences were found for the control group. Lastly, an independent t-test indicated no significant differences between the groups (Table 4).

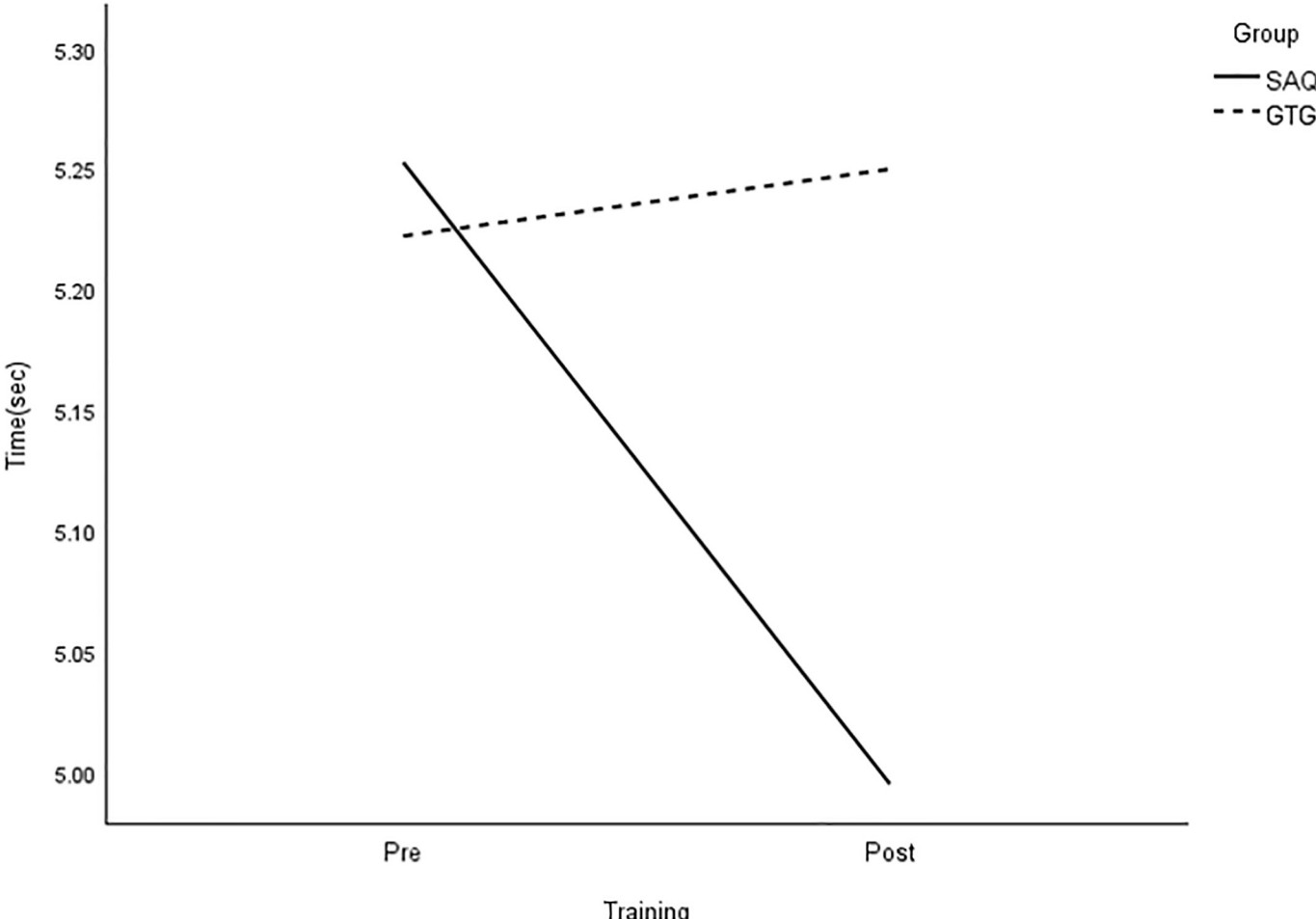

**Fig 6. Interaction plot for 30-m sprint.** *Note*: SAQ = speed, agility, and quickness training group. GTG = general training group. The results of a two-way mixed analysis of variance (ANOVA) showed a significant group × time interaction for 30-m sprint ($p < .01$) during the 8-week training period, while the main effects for time were significant for the GTG group only.

## Discussion

The purpose of the study was to evaluate the effects of implementing an 8-week traditional and ball-oriented training combined with SAQ training on high-intensity performance in U-20 female football players competing at the national level. The main findings were as that: (a) the players in the SAQ group significantly enhanced sprint speed over long distances and COD ability, compared to those in the control group, and (b) the SAQ training programme resulted in greater improvements in all tested variables from the pre-test to the post-test, except for the arrowhead agility test (left). Although some significant changes occurred between the pre-test and the post-test for the control group, the players in this group actually increased testing time in the arrowhead agility (left), arrowhead agility with dribble for both sides, and the SEMO agility, with the only exception of a decrease in 10-m sprint time. Therefore, the results of this study demonstrate the benefits of combining SAQ training with regular football practices for improving sprinting speed, COD speed, and reactive agility performance of players.

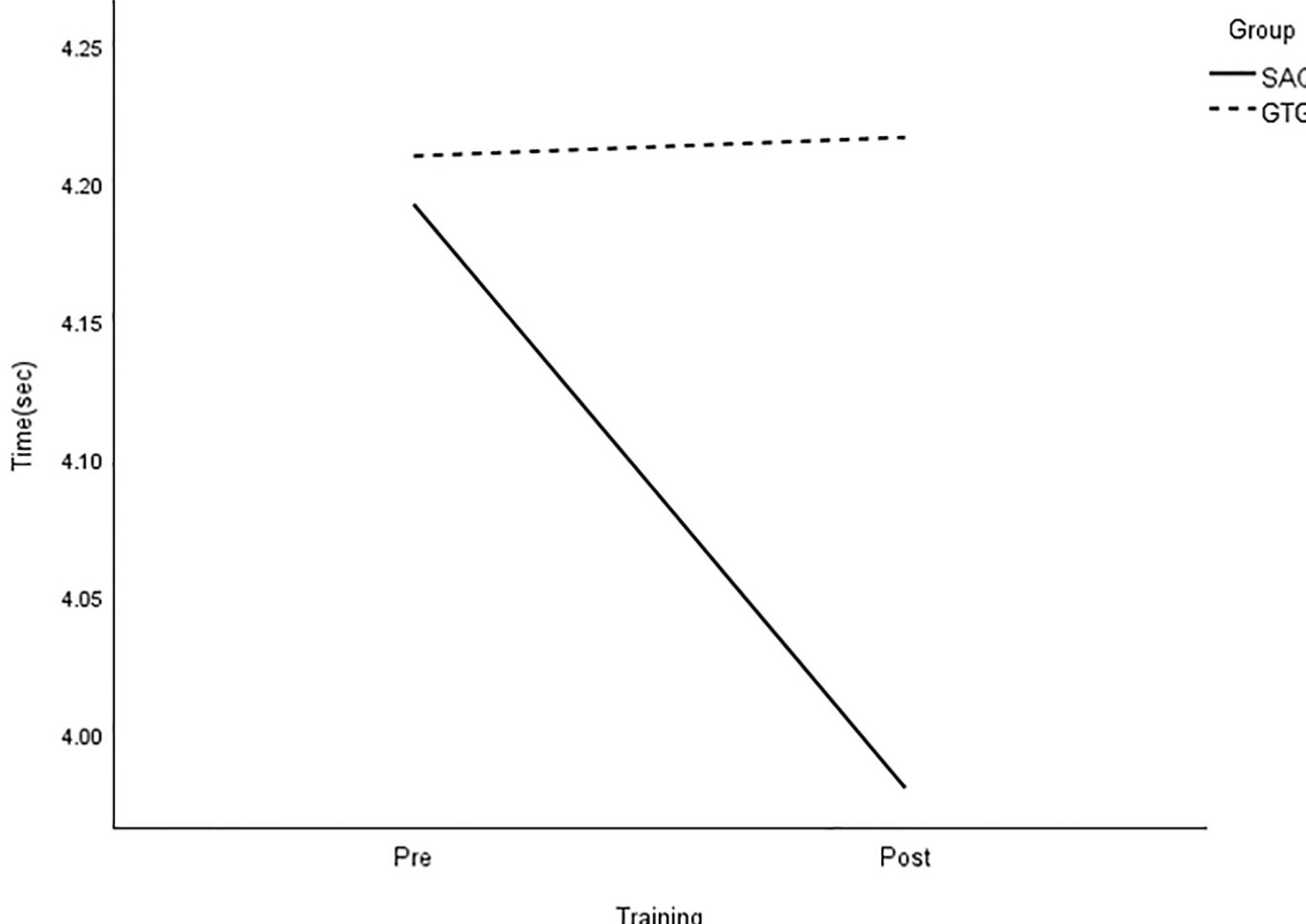

**Fig 7. Interaction plot for 20-m dribble.** *Note*: SAQ = speed, agility, and quickness training group. GTG = general training group. The results of a two-way mixed analysis of variance (ANOVA) indicated a significant group × time interaction for 20-m dribble ($p < .01$) after the 8-week training. The main effects for time were significant for the GTG group only.

Maximum sprint speed is an essential physical element for football players [15]. The results showed that only the experimental group, who underwent 8 weeks of SAQ training, significantly reduced sprint times across all speed tests. Although there were significant differences between the two groups in all speed tests at post-testing, except for the short sprints, and between the pre-post assessments in 5-, 10-, 30-m sprints without a ball and 20-m sprint with dribbling (all $p < .05$), such improvements were only observed in the SAQ group, not in the control group (Tables 3 and 4). These findings are consistent with previous research [7,12,17], which demonstrated significant enhancements in acceleration and maximum sprint speed over short distances (5- and 10-m). Similarly, the results obtained in this study provide support for our experimental hypothesis that an 8-week SAQ training method improves sprint performances over 20- and 30-m with and without dribbling. Although there is some evidence suggesting that a 12-week SAQ training significantly reduces sprint times over short [19] and long distances [20], it was unclear whether 8 weeks of an SAQ training programme would yield similar results. The results of the current study show that 8 weeks of SAQ protocol can lead to meaningful changes in acceleration speed and high-running velocity.

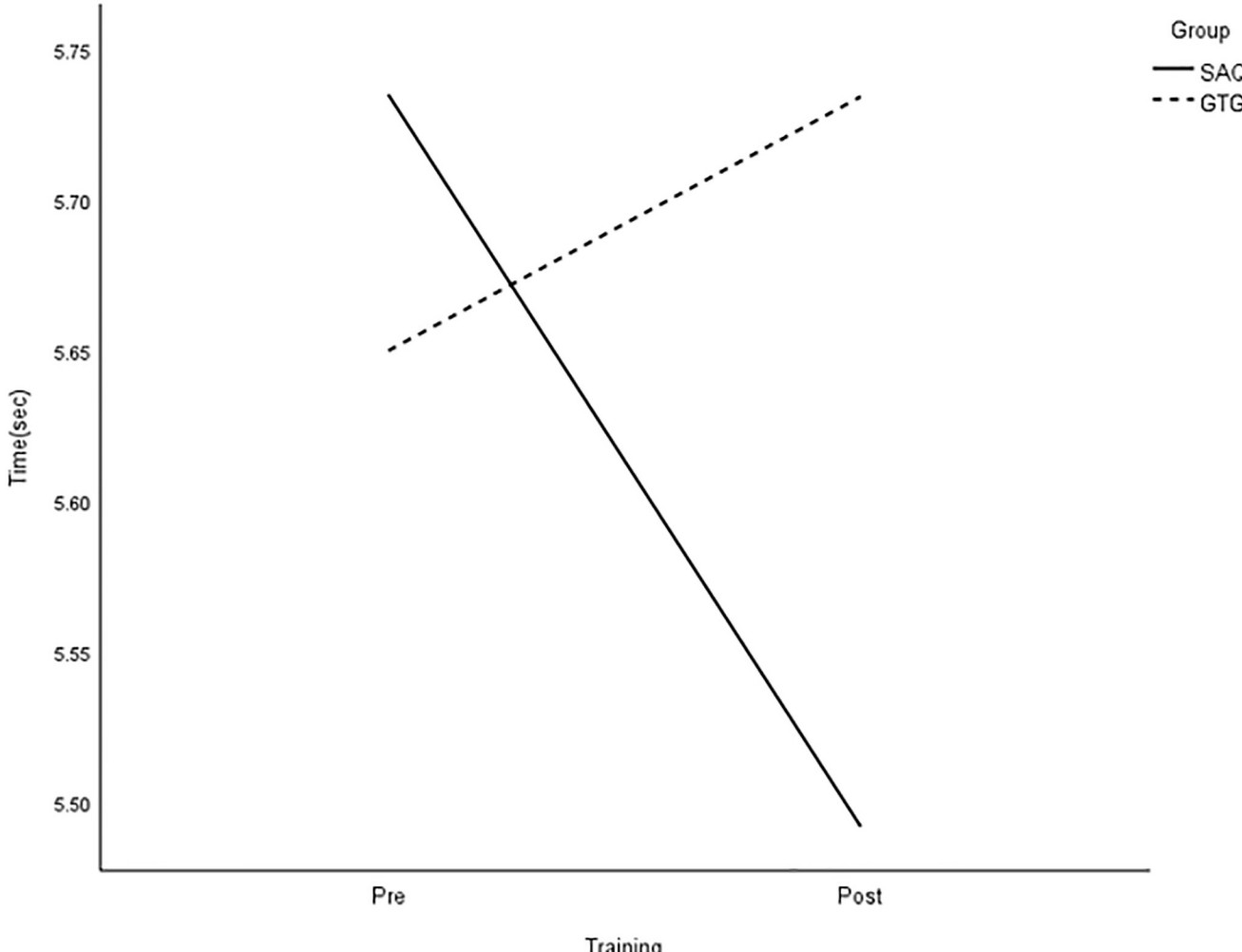

**Fig 8. Interaction plot for 30-m dribble.** *Note*: SAQ = speed, agility, and quickness training group. GTG = general training group. The results of a two-way mixed analysis of variance (ANOVA) showed a significant group × time interaction for 30-m dribble ($p < .01$) after the 8-week training period, while no main effects were found.

However, it should be noted that the control group also showed significant improvements in 10-m sprint performance after engaging in general football training. One possible explanation for the enhanced values may be due to different mechanisms at play during the 5 and 10 m sprints [17]. As suggested by Milanović et al. [7], football players are involved in high-intensity activities and directional changes that last 2–4 seconds and require short sprints (e.g., no more than 15 m). Thus, while an SAQ training protocol appeared to be effective in enhancing all speed performance in this study, further investigation is still necessary to fully understand the differences in sprint times across various distances.

COD speed is the ability of players to accelerate, decelerate, cut, and turn rapidly when changing direction. This plays an important role in optimising the high-intensity performance and reducing the risks of injuries [12]. In the present study, there were significant group × time interactions across the tests. Supporting our hypothesis, the results demonstrated greater improvements in the arrowhead agility for both sides (with and without

**Table 4. Means, standard deviations, and results of independent and paired *t*-tests for the SAQ and GTG group.**

| Variables | SAQ (*n* = 9) | | | | GTG (*n* = 10) | | | | Posttest scores | |
|---|---|---|---|---|---|---|---|---|---|---|
| | Means ± *SD* | | Paired *t*-test *df* = 8 | | Means ± *SD* | | Paired *t*-test *df* = 9 | | Independent *t*-test *df* = 17 | |
| | Pre | Post | *t* | *p(d)* | Pre | Post | *t* | *p(d)* | *t* | *p(d)* |
| Speed | | | | | | | | | | |
| 5-m sprint | 1.45 ± .08 | 1.35 ± .05 | 4.55 | .01(1.51)** | 1.41 ± .08 | 1.38 ± .07 | 1.00 | .34(.31) | -1.00 | .33(.45) |
| 10-m sprint | 2.30 ± .06 | 2.14 ± .09 | 5.12 | .01(1.70)*** | 2.30 ± .09 | 2.21 ± .04 | 2.87 | .02(.89)* | -2.08 | .06(.92) |
| 20 m sprint | 3.74 ± .14 | 3.57 ± .12 | 3.29 | .01(1.09)** | 3.67 ± .20 | 3.74 ± .17 | -1.71 | .12(.54) | -2.32 | .03(1.06)* |
| 30-m sprint | 5.25 ± .17 | 4.99 ± .15 | 5.09 | .01(1.69)*** | 5.22 ± .13 | 5.24 ± .29 | -.41 | .68(.13) | -2.29 | .03(1.05)* |
| 20-m dribble | 4.19 ± .12 | 3.97 ± .13 | 4.52 | .01(1.50)** | 4.20 ± .16 | 4.21 ± .26 | -.10 | .92(.03) | -1.11 | .02(1.11)* |
| 30-m dribble | 5.73 ± .23 | 5.49 ± .10 | 2.92 | .01(.97)** | 5.64 ± .39 | 5.73 ± .20 | -1.09 | .30(.34) | -1.42 | .01(1.42)** |
| COD Speed | | | | | | | | | | |
| Arrowhead agility (L) | 9.41 ± .24 | 9.29 ± .28 | 2.04 | .07(.68) | 9.09 ± .29 | 9.35 ± .38 | -3.01 | .01(.95)* | -0.36 | .72(.16) |
| Arrowhead agility (R) | 9.32 ± .27 | 9.10 ± .27 | 4.98 | .01(1.66)*** | 9.12 ± .21 | 9.27 ± .32 | -1.83 | .09(.58) | -1.45 | .16(.66) |
| SEMO agility | 12.16 ± .24 | 11.52 ± .36 | 5.15 | .01(1.71)*** | 11.53 ± .48 | 12.09 ± .56 | -3.79 | .01(1.20)** | -1.27 | .01(1.21)* |
| Arrowhead dribble (L) | 10.96 ± .28 | 10.70 ± .34 | 3.10 | .01(1.03)* | 11.14 ± .58 | 11.64 ± .50 | -3.34 | .01(1.05)** | -4.64 | .01(2.13)*** |
| Arrowhead dribble (R) | 10.99 ± .42 | 10.74 ± .49 | 4.31 | .01(1.43)** | 10.68 ± .44 | 11.22 ± .55 | -3.67 | .01(1.16)** | -1.96 | .06(.90) |
| 22-m slalom dribble† | 14.93 ± .89 | 13.10 ± .72 | 6.80 | .01(2.26)*** | 13.95 ± .44 | 13.95 ± .59 | .01 | .99(.01) | — | — |
| Reactive Agility | | | | | | | | | | |
| Reactive agility | 2.28 ± .86 | 2.21 ± .06 | 4.28 | .01(1.42)** | 2.28 ± .09 | 2.22 ± .16 | .81 | .43(.25) | -1.37 | .18(.63) |

Note: SAQ = speed, agility, and quickness. GTG = general training group. *n* = sample size. *SD* = standard deviation. *df* = degrees of freedom. COD = change-of-direction. SEMO = Southeast Missouri. L = left. R = right. Mdn = median. Q1 = Quartile 1. Q3 = Quartile 3.

*$p < .05$.

**$p < .01$.

***$p < .001$.

Times are expresses in second.

†Nonparametric test was performed using the Mann-Whitney test indicating that the posttest scores of the 22-m slalom dribble test were significantly lower for the SAQ group (*Mdn* = 12.88; *Q1* = 12.67, *Q3* = 13.88) than for the GTG group (*Mdn* = 13.92; *Q1* = 13.43, *Q3* = 14.53), *U* = 16, *z* = -2.36, *p* = 0.01.

dribbling), SEMO agility, and 22-m slalom dribble performances for the SAQ group after 8 weeks of training, compared to the control group, who underwent general football training. Moreover, there were significant differences between the two groups in the arrowhead agility test (left) after each programmed training, while significant decreases occurred between the pre-post assessments in 22-m slalom dribble test for the SAQ group (Table 4). Follow-up analyses, however, showed significant differences between the two groups in the SEMO, arrowhead dribble test (left), and 22-m slalom dribble tests at post-testing (Table 4). Interestingly, the control group demonstrated significant increases in the arrowhead agility (left), the SEMO, and arrowhead dribble tests. On the contrary, the SAQ training programme induced significant changes in all COD abilities for the experimental group, except for the arrowhead agility (left).

These findings support results of previous investigations [17,22] that found significant improvements in a T-test after an SAQ intervention. In a study of female basketball players, Moselhy [37] assessed COD speed using other parameters, such as the Illinois agility test while Jovanovic et al. [12] used vertical, lateral, and/or horizontal jumps to measure agility of football players, all of which showed significant enhancements in COD abilities through implementing an SAQ training programme. Amongst others (e.g., the Illinois agility test, T-test, 505 test, and

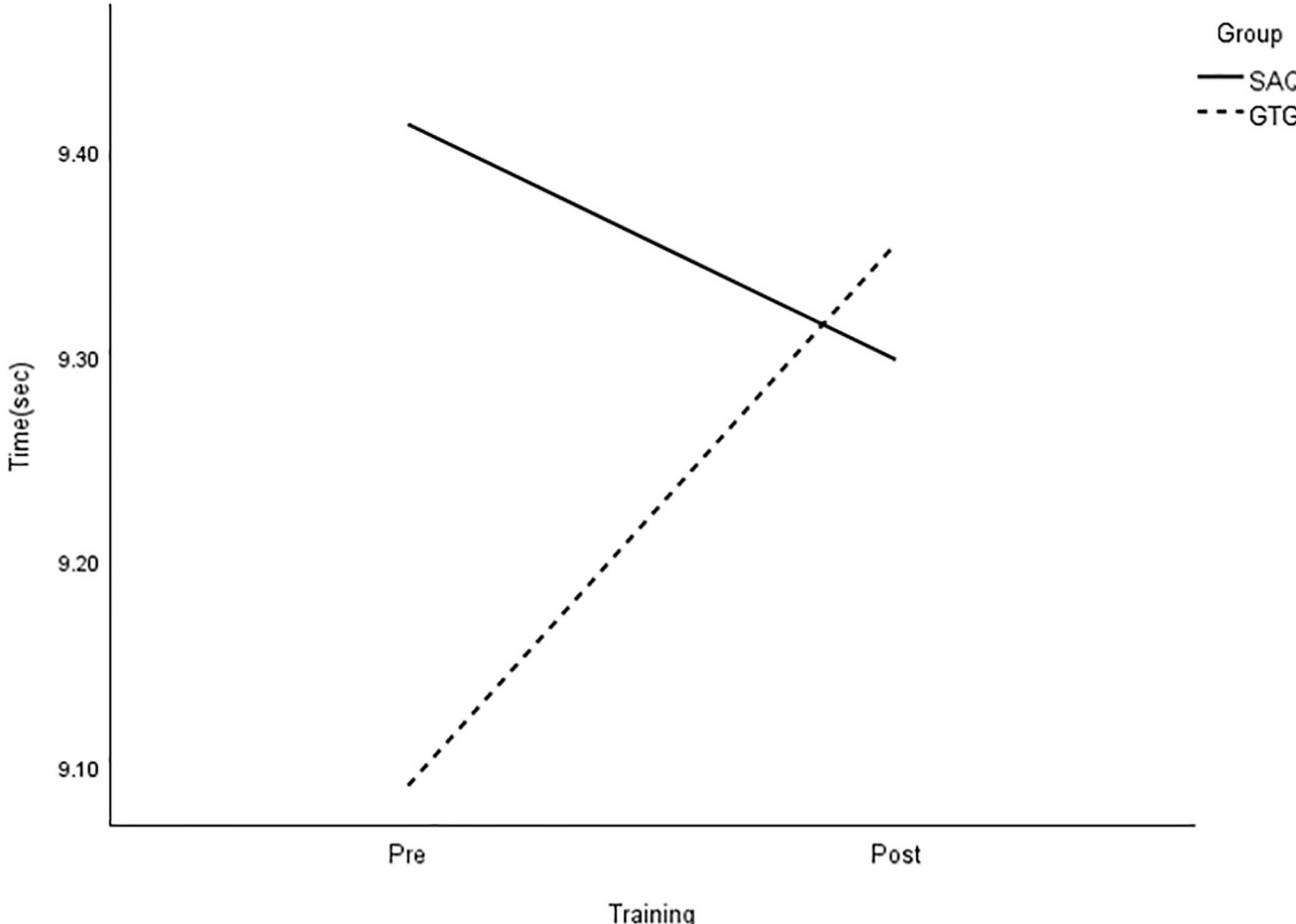

**Fig 9. Interaction plot for arrowhead agility (L).** *Note*: SAQ = speed, agility, and quickness training group. GTG = general training group. The results of a two-way mixed analysis of variance (ANOVA) showed a significant group × time interaction for the arrowhead agility test (left) after 8 weeks training ($p <$ .01). No significant main effects were observed.

Zigzag test), COD performance with and without a ball in this study was evaluated by the arrowhead agility, the SEMO, and 22-m slalom dribble tests, which are considered as reliable and valid methods for assessing pre-planned, multidirectional movements in football [32,33]. As revealed in the paired t-tests, it can be concluded that SAQ training was beneficial for improving COD speed of football players to perform faster and more agile, pre-planned, and multidirectional movements during a game.

Reactive agility represents open skills that require players to read accurately and respond quickly to situational cues to perform actions efficiently [10]. As such, visual scanning, decision-making, and reaction time are the major aspects of reactive agility, all of which are critical for information processing during a football match [20]. In this study, no significant interactions or differences between the groups were found; yet, the results were similar to previous research indicating greater improvements in reactive agility performance in the SAQ group [17]. The findings are particularly in line with previous research, which reduced time in the reactive agility test through SAQ training [20]. The pre-post differences in the reactive agility test support our experimental hypothesis that an 8-week SAQ training significantly leads to

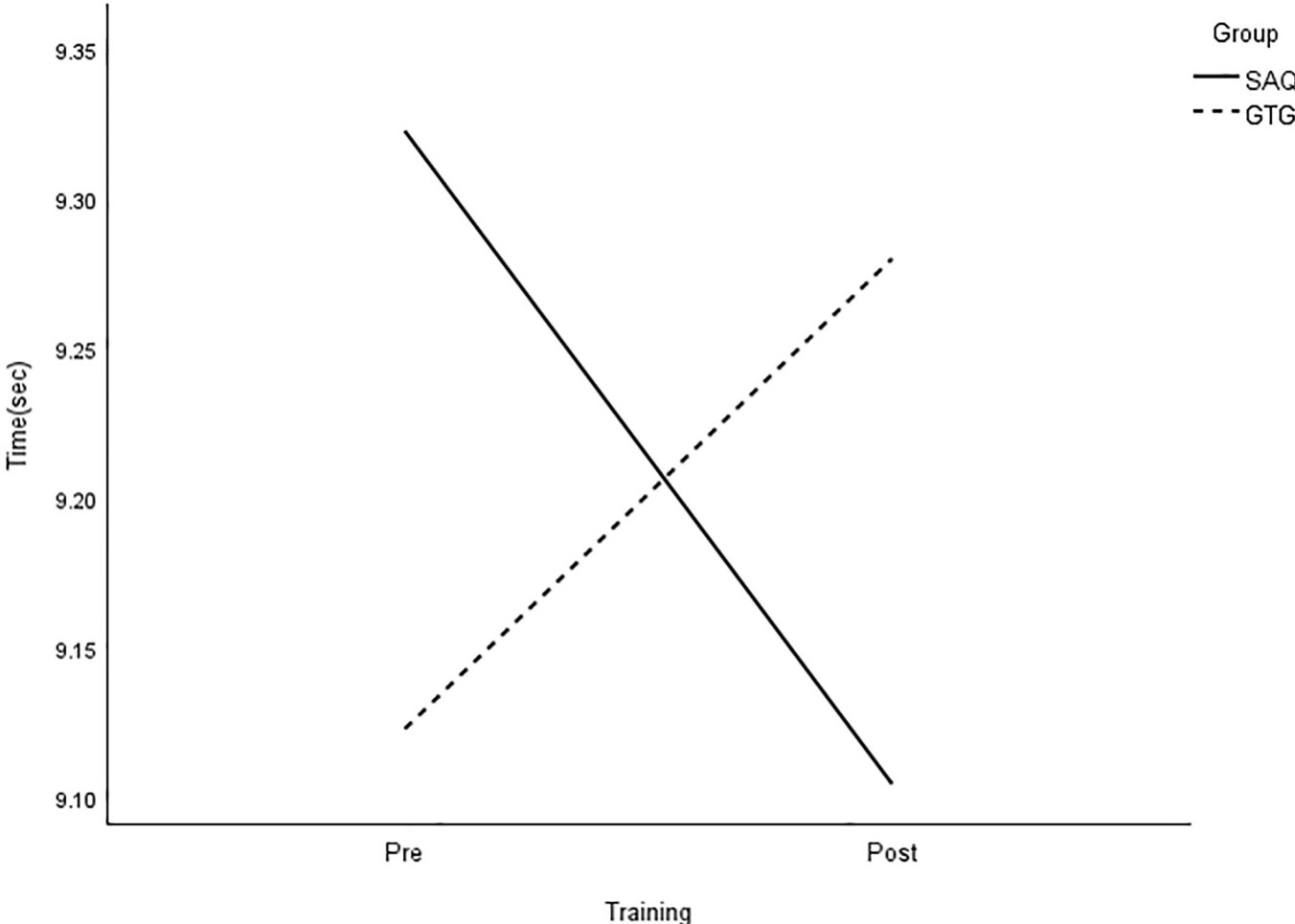

**Fig 10. Interaction plot for arrowhead agility (R).** *Note*: SAQ = speed, agility, and quickness training group. GTG = general training group. The results of a two-way mixed analysis of variance (ANOVA) indicated a significant group × time interaction for the arrowhead agility test (right) during the 8-week training period ($p < .01$), while no main effects were found. Pre to post changes were found in the SAQ group only ($p < .01$).

faster performance in the reactive agility test. Consequently, it can be suggested that SAQ training is an effective method for improving the perceptual and decision-making components of reactive agility in female football players.

## Implications, limitations, and conclusion

We conclude with a discussion of the implications and limitations of our research. The results from the present study highlight the importance and effectiveness of incorporating SAQ training into football-specific practices, particularly enhancing the high-intensity performance in U-20 female football players during the pre- and in-season. From a practical standpoint, football coaches and conditioning practitioners can implement a periodised SAQ conditioning over 8 weeks, which can enhance players' ability to sprint and perform multidirectional movements (linear, lateral, and diagonal) without losing maximal running speed and body control. By combining SAQ exercises with regular football training, significant improvements in sprint performance over short distances and COD speed can be achieved. It is worth noting that

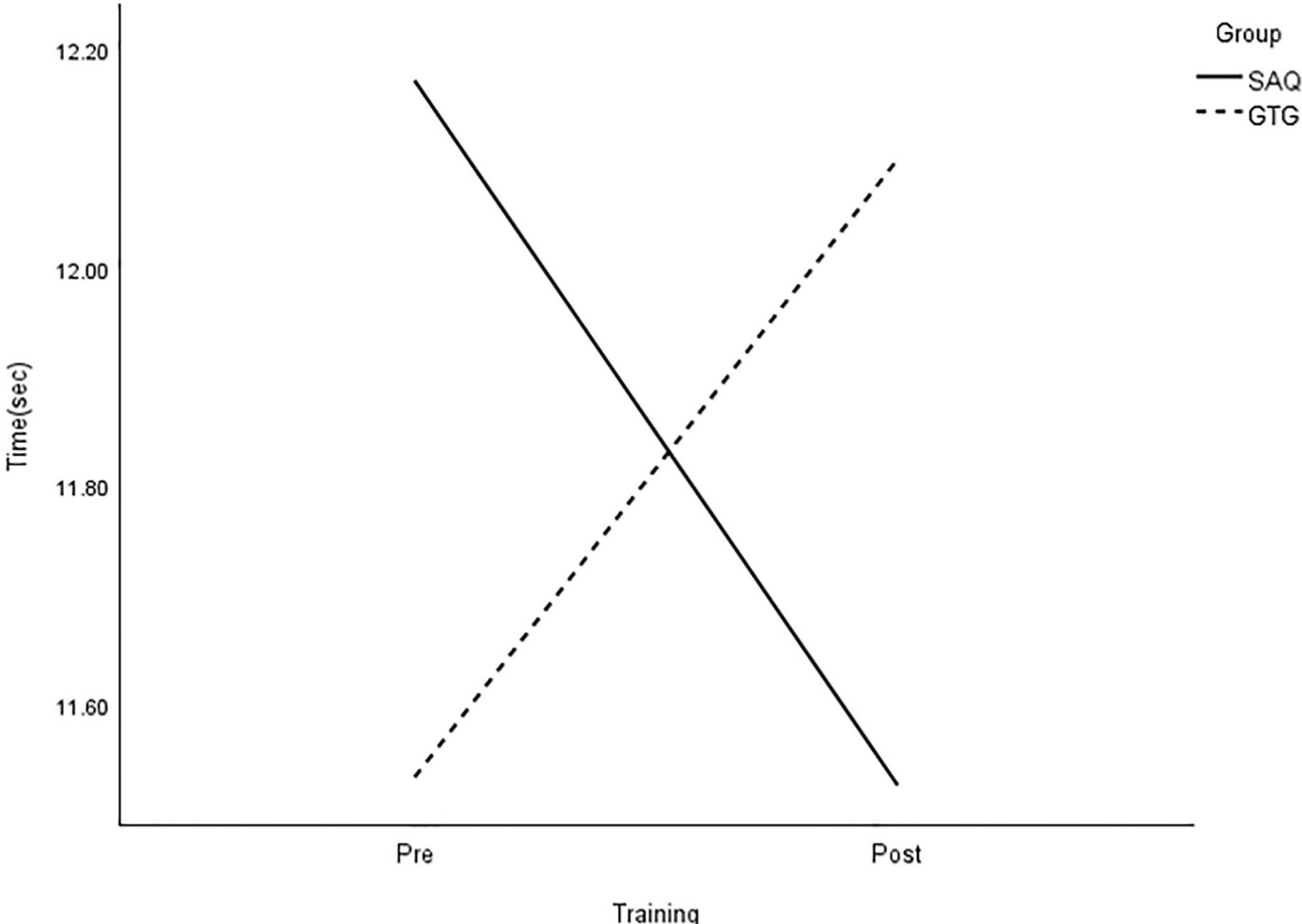

**Fig 11. Interaction plot for SEMO agility.** *Note*: SAQ = speed, agility, and quickness training group. GTG = general training group. The results of a two-way mixed analysis of variance (ANOVA) showed a significant group × time interaction for the SEMO agility test after the 8-week training ($p < .01$), while no main effects were observed.

although spring speed and COD speed are separate and independent qualities [38,39], a strong association between higher levels of sprint speed over short distances, including acceleration speed enhanced COD speed. Additionally, it is also crucial to note that acceleration, deceleration, and COD speed involve distinct techniques [40,41], where separate training and evaluation sessions should be taken to assess players' physical abilities in sprinting over short and long distances, and their agility in changing direction.

As with all research, our study is subject to a number of limitations that can represent avenues for future research. First, we focused on female youth football players who compete at the national level. Thus, generalisation of the results to other populations in different team sports (rugby or hockey), performance levels (elite or amateur) or age groups (U-12 or U-15) should be done with caution due to the different nature of various sports and variations in physical demands. Considering individual variations and specific player characteristics, future research can explore the applicability of SAQ training to different age groups and sports and investigate its effects on sprint and agility performance. Further, our sample

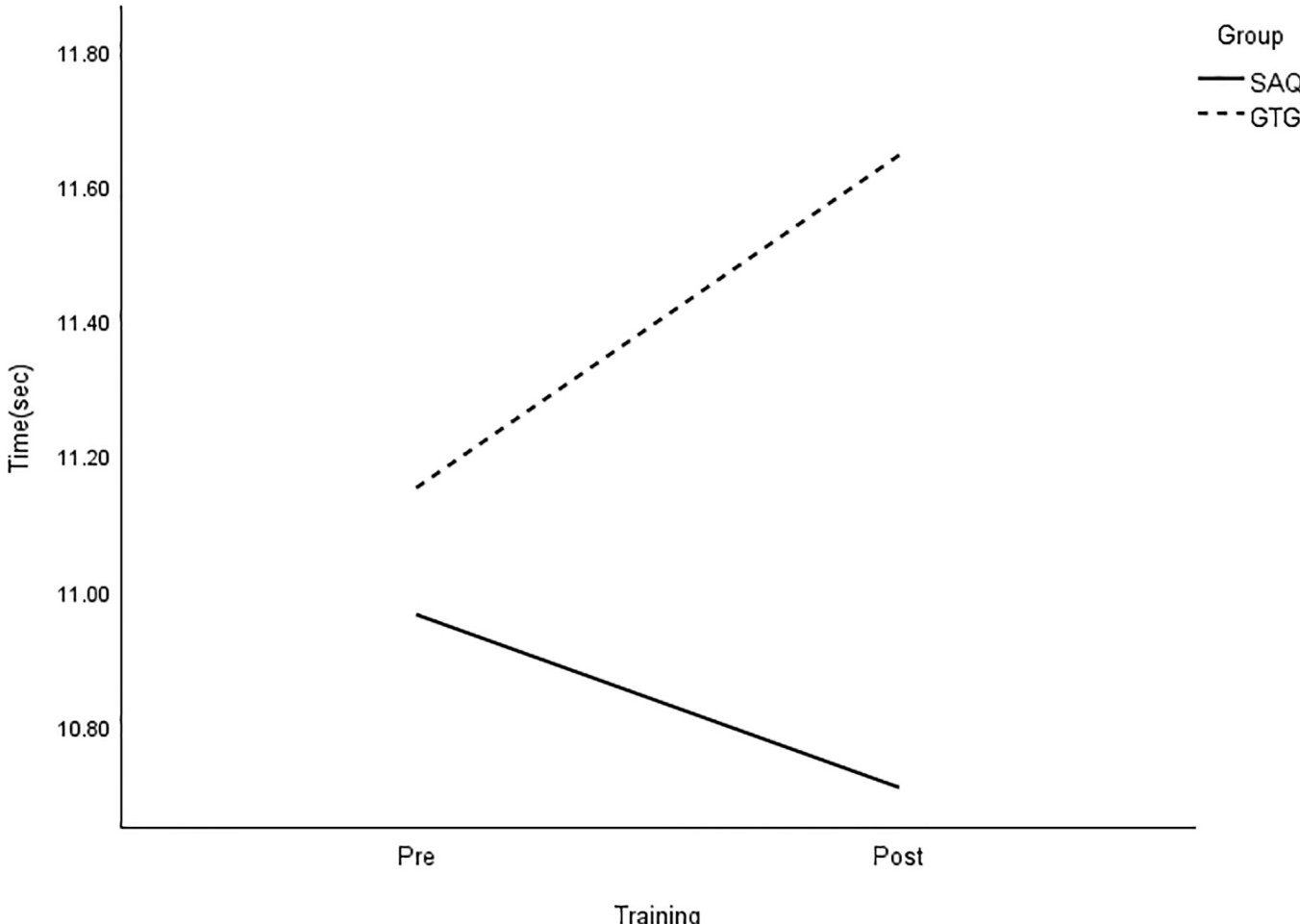

**Fig 12. Interaction plot for arrowhead dribble (L).** *Note*: SAQ = speed, agility, and quickness training group. GTG = general training group. The results of a two-way mixed analysis of variance (ANOVA) showed a significant group × time interaction for the arrowhead dribble test (left; $p < .01$), while the main effects for group were found, indicating a different mean change between the SAQ and GTG groups.

represents a specific group of female footballers in East Asia, and the study was cross-sectional, with the small sample size. Indeed, the scarcity of highly trained female football players presents challenges in achieving a large sample size of participants with the desired level of training. Thus, the results may not be generalisable to other geographic settings and time periods. To remedy this shortcoming and increase the potential for generalisability, we encourage future researchers to conduct additional trials in diverse geographical contexts and sporting settings.

We would also like to acknowledge the inherent limitations of using a stopwatch for timing. Despite our effort to minimise potential impacts on the reliability of results, the use of a stopwatch can introduce some variability, even with the same rater. Hence, future researchers could explore alternative methods for timing, such as electronic timing systems or motion capture technology (e.g., speed gate). Doing so will provide more precise measurements. Furthermore, HR and RPE were consistently monitored as indicators of training intensity. However, the lack of documentation and recording of these indicators can be considered a limitation of

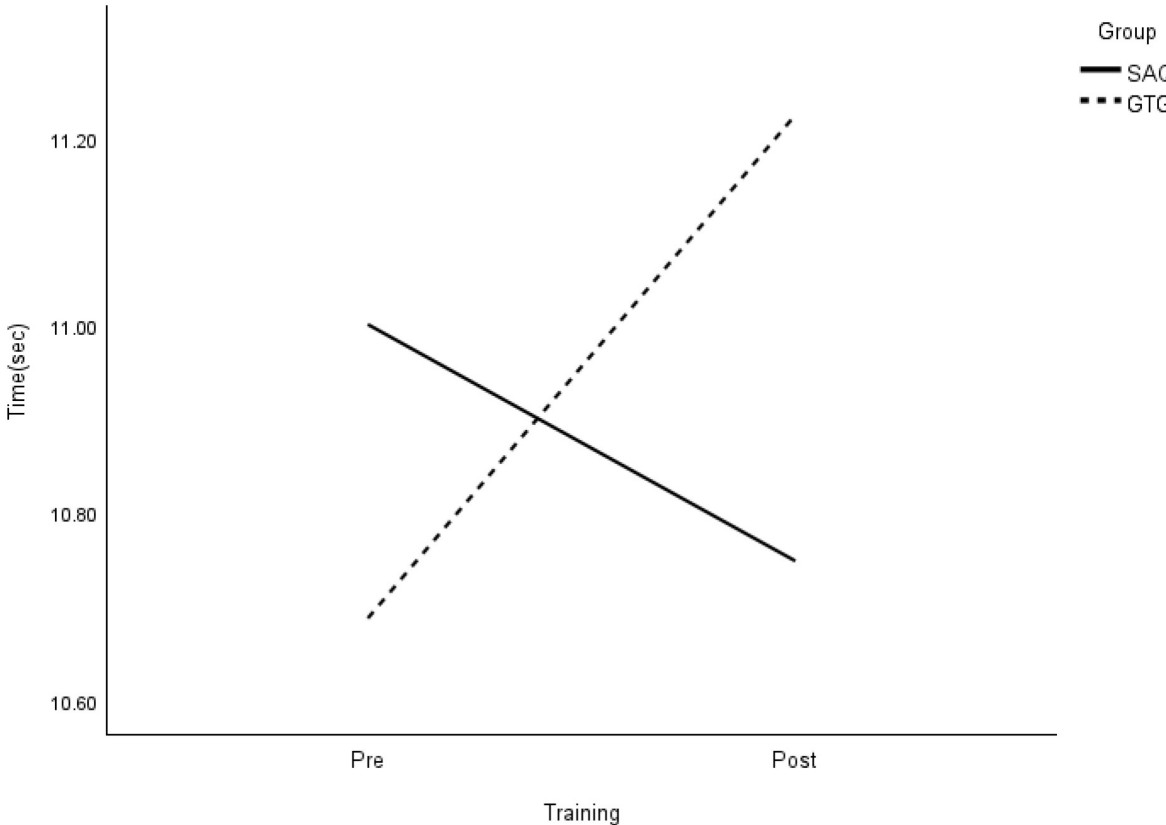

**Fig 13. Interaction plot for arrowhead dribble (R).** *Note*: SAQ = speed, agility, and quickness training group. GTG = general training group. The results of a two-way mixed analysis of variance (ANOVA) indicated a significant group × time interaction for the arrowhead dribble test (right) after 8 weeks training ($p < .01$), while no main effects were found. The significant main effects of time and group were also found. Specifically, there was a significant improvement in COD speed in the 22-m slalom dribbling test (Fig 14), and another significant difference between groups in the arrowhead dribble (left).

the study. These measures are commonly used to assess exercise intensity, and their absence may limit the comprehensive understanding of the participants' physiological responses during training. Additionally, we note that the validity of the arrowhead test has not been specifically tested in previous studies [32,33], although its reliability has been reported. Similarly, the SEMO agility test has not been assessed for validity and reliability, but it is widely used in football, while dribbling and slalom tests have been found to be both valid and reliable [42]. In light of the current findings, researchers could focus on evaluating the validity and reliability of both the arrowhead and SEMO tests. Finally, as the present study indicates, coaches and conditioning practitioners can implement SAQ exercises 2–3 times a week to transfer training effects to sprint performance over short and long distances, as well as COD abilities of players. Enhanced acceleration, sprint speed, COD ability, and reactive agility can contribute to better athletic performance in high-intensity activities and ballistic and dynamic movements during a football game [35]. Together, as shown in this study, an SAQ training programme with a gradual increase in intensity is more effective for improving physical capacities committed to high-intensity performance than regular football training alone. In summary, 8 weeks of SAQ training appear to induce substantial changes in sprint and COD speed in highly trained female youth football players.

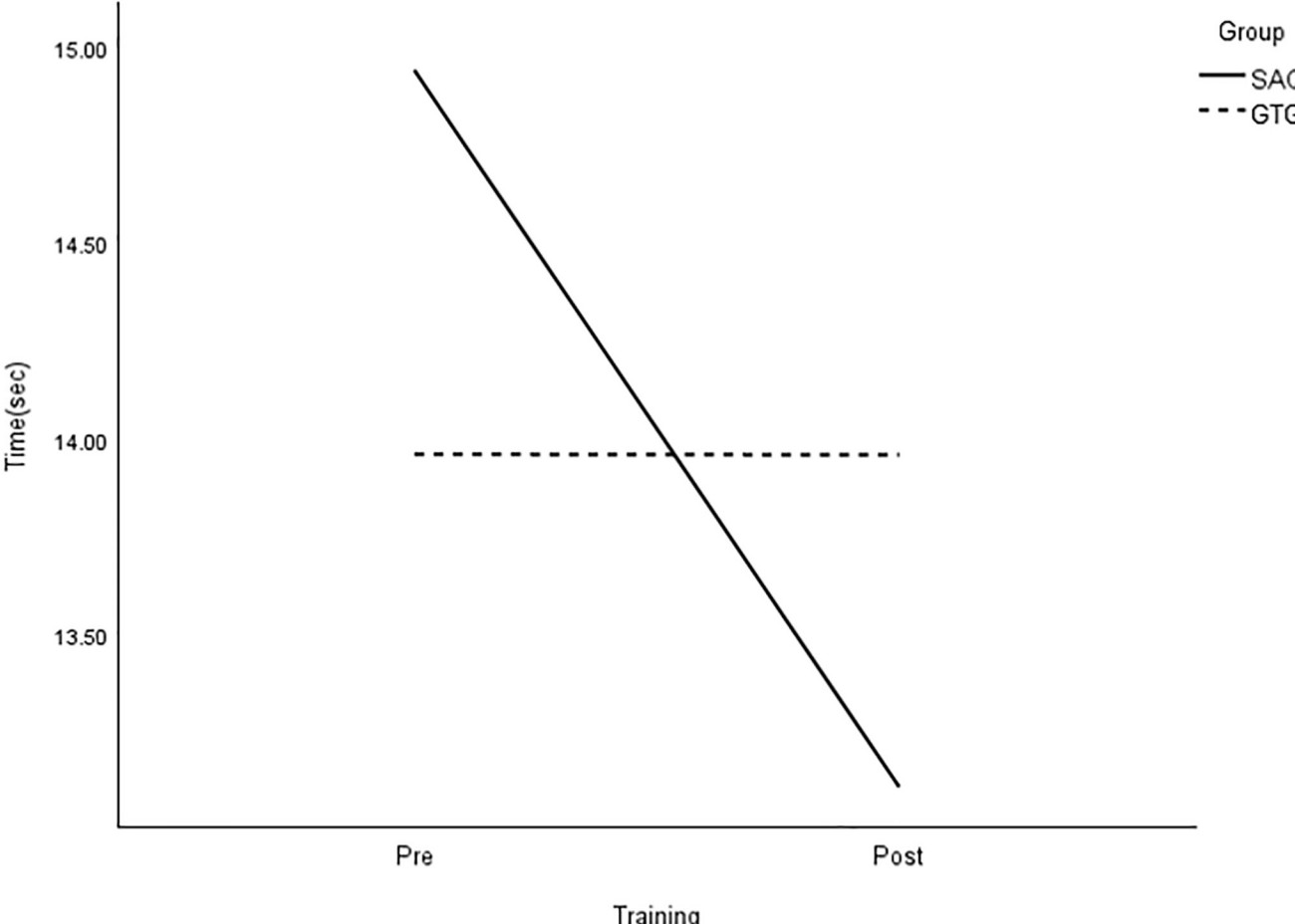

**Fig 14. Interaction plot for 22-m slalom dribble.** *Note*: SAQ = speed, agility, and quickness training group. GTG = general training group. The results of a two-way mixed analysis of variance (ANOVA) indicated a significant group × time interaction for 20-m dribble ($p < .01$) after the 8-week training. The main effects for time were significant for the GTG group only.

## Supporting information

**S1 File.**
(CSV)

## Acknowledgments

The author(s) would like to acknowledge all the participants for their contributions to the study. The author(s) declare no conflicts of interest associated with this publication, and there is no specific funding for this work.

## Author Contributions

**Conceptualization:** Young-Soo Lee.

**Data curation:** Dayoung Lee.

**Formal analysis:** Na Young Ahn.

**Investigation:** Dayoung Lee.

**Methodology:** Young-Soo Lee, Dayoung Lee, Na Young Ahn.

**Project administration:** Dayoung Lee.

**Resources:** Dayoung Lee.

**Software:** Na Young Ahn.

**Supervision:** Young-Soo Lee.

**Validation:** Dayoung Lee.

**Visualization:** Na Young Ahn.

**Writing – review & editing:** Young-Soo Lee, Dayoung Lee, Na Young Ahn.

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
