## [Decision Letter · Decision Letter 0]

16 May 2022

PONE-D-22-09664SAQ training on sprint, change-of-direction speed, and agility in elite U-20 female football playersPLOS ONE

Dear Dr. Lee,

Thank you for submitting your manuscript to PLOS ONE. After careful consideration, we feel that it has merit but does not fully meet PLOS ONE’s publication criteria as it currently stands. Therefore, we invite you to submit a revised version of the manuscript that addresses the points raised during the review process.

Please, address point-to-point all reviewers' (especially Reviewer 1's and 3's) issues.

We look forward to receiving your revised manuscript.

Kind regards,

Luca Paolo Ardigò, Ph.D.

Academic Editor

PLOS ONE

Journal Requirements:

 [Unfunded studies]. 

[No competing interests].

6. We note that you have referenced (ie. Bewick et al. [5]) which has currently not yet been accepted for publication. Please remove this from your References and amend this to state in the body of your manuscript: (ie “Bewick et al. [Unpublished]”) as detailed online in our guide for authors

http://journals.plos.org/plosone/s/submission-guidelines#loc-reference-style.

7. Please include your tables as part of your main manuscript and remove the individual files. Please note that supplementary tables should be uploaded as separate "supporting information" files.

Additional Editor Comments:

Please, address point-to-point all reviewers' (especially Reviewer 1's and 3's) issues.

Reviewers' comments:

Reviewer's Responses to Questions

**Comments to the Author**

1. Is the manuscript technically sound, and do the data support the conclusions?

Reviewer #1: No

Reviewer #2: Yes

Reviewer #3: Yes

2. Has the statistical analysis been performed appropriately and rigorously? 

Reviewer #1: No

Reviewer #2: Yes

Reviewer #3: Yes

3. Have the authors made all data underlying the findings in their manuscript fully available?

Reviewer #1: Yes

Reviewer #2: No

Reviewer #3: Yes

4. Is the manuscript presented in an intelligible fashion and written in standard English?

Reviewer #1: Yes

Reviewer #2: Yes

Reviewer #3: No

5. Review Comments to the Author

Reviewer #1: First, I would like to commend the authors for doing research in an area where scientists barely have scratched the surface – namely women’s football. However, after a thorough review, it is my opinion that the experiment and statistics in the paper do not meet the technical standards set by the journal criteria.

Under you will find some specific concerns of mine:

Line 100: The description of the participants does not convince me that they are part of an “elite” population. They could, however, be part of an “top-level” population. I would further recommend reading the paper “Defining Training and Performance Caliber: A Participant Classification Framework” by McKay et al (2022).

Line 102: The sample size calculation does not seem to be correct, and the authors do not specify which statistical test they did the analysis for. Using G*Power software version 3.1.9.7 for a difference between two independent means, with an effect size of 0.3, power of 80%, and alpha at 0.05, I end up with a sample size of 139 in each group! This is very different from the stated minimum sample size of 16. That said, achieving such a sample size (139 in each group) given the population (top-level) is unrealistic. If I could make a suggestion, I would recommend that the authors switch to a Bayesian framework instead.

Line 110: How the players where randomized is never stated.

Line 142-145: It is stated that SAQ training was performed with a progressive intensity from 80% to 100%. But you do not state how you measured intensity. Was it RPE or something else?

Testing protocol: The speed testing is both nicely done and described in detail. However, I take issue with the inclusion and execution of the COD and agility tests. First, I looked through the stated references for the reliability of the COD tests (30-32) and the reactive agility test (20). I could not see that these papers set out to determine the reliability of said tests. Furthermore, I would also imagine using a stopwatch would make the timing unreliable even if it is the same rater. Finally, most of the time in the COD and agility tests are spent running instead of changing directions or reacting. It is therefore questionable if these tests are valid and reliable measures of COD or agility. I would also suggest reading some of the literature by Sophia Nimphius on the subject.

Line 243-245: It is first mentioned that pairwise comparisons were not applied because the factors entailed only two groups. This is correct. However, you then go on to say that Bonferroni correction was conducted to determine any significant differences between the two groups or test occasions for each group. This does not make any sense. Also, there is no need to do follow-up independent and paired t-test information in a mixed 2x2 ANOVA – which is actually what you first imply. All the information needed is in the interaction term.

Line 254: Perhaps nitpicky, but you should state whether you used the pre-test and post-scores, or the change scores, to determine normality. The former is incorrect, while the latter is correct.

Line 255: I suspect you did an independent t-test at baseline to check that the randomization procedure worked correctly. However, one should actually never do this. The more tests you do the larger probability that one of the tests turns is significant purely by chance.

Results section: As mentioned previously, the independent and paired t-test are redundant and should be removed. Instead, I would suggest that you report the mean change ± 95% confidence intervals for both groups, and the mean difference ± 95% confidence intervals between the groups.

Reviewer #2: The aim of the present study was to investigate the effect of an 8-week speed, agility, and quickness (SAQ) training on linear sprint speed, change of-direction speed, and reactive agility in elite U-20 female football players. The study is well written, easy to read, and provides new insights into the potentiality of SAQ training for improving various physical features of soccer players. Introduction is clear and provides an appropriate description of background and rationale. Methods are well described. Results section can be improved in terms of its readability (although I understand that describing results of two way anova may not be easy). Discussion provides a depth analysis of results, in light of the literature on SAQ methodology. I would congratulate the Authors for their work. I have no particular comments, apart from minor ones that I hope may be useful to improve the overall quality of the manuscript.

Line 47-49. The contrary? Maximum speed 10-30 m; acceleration 0 to 5 m.

Line 152. Were the players familiarized with the testing protocols?

Table 3 is appropriate. However, I would suggest to consider to substitute the table with graphs, that would allow an immediate visual impression of the behaviour of the variables tested for the two groups. I understand that including graphs for all the variables may be impractical, but graphs for some variables (the most important ones) may be useful. Moreover, please insert units of measurements for the variables in the table (for examples speed).

Reviewer #3: The aim of the study was investigating the effects of a relatively long training program based on SAQ on physical performance in elite female football players.

The manuscript cover an interesting topic, providing additional information on the usefulness of SAQ for improving performance in a female sample of soccer players.

However, I would suggest the authors to improve the readability of the text by better proofreading the English language. Moreover, they are requested to better improve the methodological aspects in order to support what have been done. Lastly, they are also recommended to better explore the literature reporting a certain specificity between COD performance and acceleration. IMO, the completeness of the discussion section would benefit from such exploration.

Here below some specific comment

Specific comments

Abstract

line 17: suggested rewording "the effects of an 8-week speed, agility, and quickness (SAQ) training on..."

line 21: "The players were tested for:"

Intro

lines 37-38: "...actions (e.g., passing kicking, trapping, dribbling, tackling) and without ball (...)"

line 49: (0 to 5 m)

line 52: reactive is redundant. The term agility already embodies cognitive skills in which reaction plays a role.

lines 76-78: please rephrase for clarity

line 86-87: please rephrase for clarity. Perhaps also replace "researchers know little" to "little is known..."

M&M

lines 106-109: state clearly all inclusion and exclusion criteria

line 133: remove "were"

line 134: I would move this sentence from here. Outside the context.

line 135: "normal dietary"? Please be specific. Perhaps the following ref may serve to provide more detail abut that statement.

doi: 10.1080/15438627.2020.1809410

line 145: how did the authors control the intensity?

line 171: please add reference for such a choice

Results

I would suggest the authors to first stating the outcomes of the interactions.

Discussion

In the first part of this section, IMO, It is important to clarify whether the study hypothesis was verified.

lines 364-366:

this should be moved at the beginning of the introduction section, after the main findings are stated.

line 381: What about the studies suggesting that acceleration is independent of COD performance? Please consider the works of Nimphius and other authors, for example.

doi: 10.1519/JSC.0000000000001421

DOI: 10.1519/SSC.0000000000000309

doi: 10.7717/peerj.9486

6. PLOS authors have the option to publish the peer review history of their article (what does this mean?). If published, this will include your full peer review and any attached files.

Reviewer #1: No

Reviewer #2: No

Reviewer #3: No

---

## [Author Response · Author response to Decision Letter 0]

14 Oct 2023

Please see the attached responses to the reviewers' comments for more details.

Meanwhile, we would like to thank the editors for allowing us to submit the revised draft of the manuscript and the reviewers for providing us with constructive and insightful comments. Your suggestions were useful to improve the current manuscript. We studied the feedback carefully to address the reviewers’ concerns and worked on to strengthen those areas to reflect most of the comments by the reviewers. Please see the point-by-point descriptions of every single change we made in the right column below (next to the actioned one), and we marked the edits in the manuscript using the track changes function in Word.

---

## [Decision Letter · Decision Letter 1]

24 Nov 2023

PONE-D-22-09664R1SAQ training on sprint, change-of-direction speed, and agility in U-20 female football playersPLOS ONE

Dear Dr. Lee,

Thank you for submitting your manuscript to PLOS ONE. After careful consideration, we feel that it has merit but does not fully meet PLOS ONE’s publication criteria as it currently stands. Therefore, we invite you to submit a revised version of the manuscript that addresses the points raised during the review process.

Please, one further effort. Address point-to-point Reviewer 1 and 3' issues.

We look forward to receiving your revised manuscript.

Kind regards,

Luca Paolo Ardigò, Ph.D.

Academic Editor

PLOS ONE

Journal Requirements:

Additional Editor Comments:

Please, one further effort. Address point-to-point Reviewer 1 and 3' issues.

Reviewers' comments:

Reviewer's Responses to Questions

**Comments to the Author**

1. If the authors have adequately addressed your comments raised in a previous round of review and you feel that this manuscript is now acceptable for publication, you may indicate that here to bypass the “Comments to the Author” section, enter your conflict of interest statement in the “Confidential to Editor” section, and submit your "Accept" recommendation.

Reviewer #1: All comments have been addressed

Reviewer #2: All comments have been addressed

Reviewer #3: All comments have been addressed

2. Is the manuscript technically sound, and do the data support the conclusions?

Reviewer #1: Yes

Reviewer #2: Yes

Reviewer #3: Yes

3. Has the statistical analysis been performed appropriately and rigorously? 

Reviewer #1: Yes

Reviewer #2: Yes

Reviewer #3: I Don't Know

4. Have the authors made all data underlying the findings in their manuscript fully available?

Reviewer #1: Yes

Reviewer #2: Yes

Reviewer #3: Yes

5. Is the manuscript presented in an intelligible fashion and written in standard English?

Reviewer #1: Yes

Reviewer #2: Yes

Reviewer #3: Yes

6. Review Comments to the Author

Reviewer #1: I think the authors have done a superb job in updating the manuscript, which is almost ready for publication.

I only have a few comments:

1) I recommend changing the interaction plot to a pre-post dot plot so that we could see each data point. For an example see figure 4.1 in https://mladenjovanovic.github.io/bmbstats-book/causal-inference.html.

2) Could you write a sentence or two in the manuscript of how intensity was evaluated? You noted in a previous comment that RPE and Heart Rate were measured, but out of the scope of the paper and thus not reported, which is fine. However, some explanation of how intensity was evaluated and communicated to the athletes should appear in the text.

3) I'm sorry, but I fail to see how the references (30-31) underscore the validity and reliability of the tests that have been used:

Wing, C. E., Turner, A. N., & Bishop, C. J. (2020). Importance of strength and power on key performance indicators in elite youth soccer. The Journal of Strength & Conditioning Research, 34(7), 2006-2014.

Joo, C. H. (2016). The effects of short-term detraining on exercise performance in soccer players. Journal of exercise rehabilitation, 12(1), 54.

As far as I know, the sprint tests along with the dribbling/slalom tests are both valid and reliable (see: Altmann, S., Ringhof, S., Neumann, R., Woll, A., & Rumpf, M. C. (2019). Validity and reliability of speed tests used in soccer: A systematic review. PloS one, 14(8), e0220982.). Arrowhead is reliable but it's validity has not been tested. SEMO has neither been tested for validity and reliability, but I can agree that it is widely used. Please address this in the text.

4) This might be semantics on my part, but I don't like the arrowhead and SEMO to be defined as COD or agility tests. I think they are more tests of maneuverability, as they can't really isolate the COD or agility component. I would be happy if this is addressed in the text, but feel free to leave the definitions as is.

Reviewer #2: (No Response)

Reviewer #3: The authors have put a lot of effort in addressing the Reviewer's point. The manuscript has considerably improved. However, I feel that some important recent references are still missing. Please see the following:

doi: 10.1371/journal.pone.0277683

doi: 10.3390/ijerph18041962

These would help the authors to reinforce both the background and rationale of the study.

7. PLOS authors have the option to publish the peer review history of their article (what does this mean?). If published, this will include your full peer review and any attached files.

Reviewer #1: No

Reviewer #2: No

Reviewer #3: No

---

## [Author Response · Author response to Decision Letter 1]

26 Jan 2024

Overall comments: We express our gratitude to the editors for allowing us to submit our revised draft of the manuscript and to the reviewers for their valuable and constructive comments. Your suggestions have greatly contributed to the enhancement of the current manuscript. We carefully analysed the feedback to address the concerns raised by the reviewers and worked to strengthen those aspects. The right column below provides a point-by-point description of each change we made (next to the actioned one), and we have marked the edits in the manuscript using the track changes function in Word.

---

## [Decision Letter · Decision Letter 2]

6 Feb 2024

SAQ training on sprint, change-of-direction speed, and agility in U-20 female football players

PONE-D-22-09664R2

Dear Dr. Lee,

We’re pleased to inform you that your manuscript has been judged scientifically suitable for publication and will be formally accepted for publication once it meets all outstanding technical requirements.

Kind regards,

Luca Paolo Ardigò, Ph.D.

Academic Editor

PLOS ONE

Additional Editor Comments (optional):

Congratulations for the interesting work.

Reviewers' comments:

Reviewer's Responses to Questions

**Comments to the Author**

1. If the authors have adequately addressed your comments raised in a previous round of review and you feel that this manuscript is now acceptable for publication, you may indicate that here to bypass the “Comments to the Author” section, enter your conflict of interest statement in the “Confidential to Editor” section, and submit your "Accept" recommendation.

Reviewer #1: All comments have been addressed

Reviewer #3: All comments have been addressed

2. Is the manuscript technically sound, and do the data support the conclusions?

Reviewer #1: Yes

Reviewer #3: Yes

3. Has the statistical analysis been performed appropriately and rigorously? 

Reviewer #1: Yes

Reviewer #3: I Don't Know

4. Have the authors made all data underlying the findings in their manuscript fully available?

Reviewer #1: Yes

Reviewer #3: Yes

5. Is the manuscript presented in an intelligible fashion and written in standard English?

Reviewer #1: Yes

Reviewer #3: Yes

6. Review Comments to the Author

Reviewer #1: The authors have done a great job in addressing my and the other reviewers' comments. While I still think the addition of dot plots would have been more visually pleasing, this paper is ready for publication.

Reviewer #3: (No Response)

7. PLOS authors have the option to publish the peer review history of their article (what does this mean?). If published, this will include your full peer review and any attached files.

Reviewer #1: No

Reviewer #3: No

---

## [Editor Report · Acceptance letter]

21 Feb 2024

PONE-D-22-09664R2 

PLOS ONE

Dear Dr. Lee, 

I'm pleased to inform you that your manuscript has been deemed suitable for publication in PLOS ONE. Congratulations! Your manuscript is now being handed over to our production team.

Kind regards, 

on behalf of

Dr. Luca Paolo Ardigò 

Academic Editor

PLOS ONE